# Bioadhesive interface for marine sensors on diverse soft fragile species

Camilo Duque Londono [1,8], Seth F. Cones[2,8], Jue Deng [1,6,8], Jingjing Wu [1], Hyunwoo Yuk[1,7], David E. Guza[3], T. Aran Mooney [4] ✉ & Xuanhe Zhao [1,5] ✉

Marine animals equipped with sensors provide vital information for understanding their ecophysiology and collect oceanographic data on climate change and for resource management. Existing methods for attaching sensors to marine animals mostly rely on invasive physical anchors, suction cups, and rigid glues. These methods can suffer from limitations, particularly for adhering to soft fragile marine species such as squid and jellyfish, including slow complex operations, unreliable fixation, tissue trauma, and behavior changes of the animals. However, soft fragile marine species constitute a significant portion of ocean biomass (>38.3 teragrams of carbon) and global commercial fisheries. Here we introduce a soft hydrogel-based bioadhesive interface for marine sensors that can provide rapid (time <22 s), robust (interfacial toughness >160 J m$^{-2}$), and non-invasive adhesion on various marine animals. Reliable and rapid adhesion enables large-scale, multi-animal sensor deployments to study biomechanics, collective behaviors, interspecific interactions, and concurrent multi-species activity. These findings provide a promising method to expand a burgeoning research field of marine biosensing from large marine mammals and fishes to small, soft, and fragile marine animals.

Marine animal-borne sensors serve as an autonomous platform to study movements and behaviors (i.e., predator-prey interactions, habitat, and migration) of marine animals, as well as to collect oceanographic data (i.e., light, temperature, salinity, and oxygen) over large spatial and temporal scales[1–4]. These data have provided vital insights for understanding marine ecosystems[5] and aided the effective management of fisheries[6]. Electronic sensors such as biotelemetry tags that transmit data to satellites and biologging tags that store data internally have been attached to robust marine vertebrates[7] (e.g., marine mammals, seabirds, reptiles, and large fishes) via subcutaneous anchors, suction cups, and/or rigid glues. However, these sizeable and rigid sensors are not applicable for attaching to more fragile marine

organisms that may be small or have soft outer protective layers. These marine organisms include many invertebrates such as squid and jellyfish and many small fishes. Soft-bodied marine organisms are highly diverse, abundant (>38.3 teragrams of carbon[8]), and pervasive in all ocean habitats, as well as provide a key protein source for humans worldwide[9] and sequester greenhouse gasses into the deep ocean[10]. While advances in miniaturized biologging tags show the potential to monitor movements and behaviors of individual small invertebrate animals with high temporal resolution[11], physically anchoring or gluing biologging tags on soft, fragile species is invasive, slow, or difficult to properly attach[4]. Further, the resultant tissue trauma on the epidermal and muscular layers can disturb animal behaviors, have ethical

[1]Department of Mechanical Engineering, Massachusetts Institute of Technology, Cambridge, MA, USA. [2]Massachusetts Institute of Technology and Woods Hole Oceanographic Institution Joint Program in Oceanography/Applied Ocean Science & Engineering, Cambridge, MA 02139, USA. [3]Applied Engineering Solutions LLC, Columbus, OH, USA. [4]Biology Department, Woods Hole Oceanographic Institution, Woods Hole, MA 02543, USA. [5]Department of Civil and Environmental Engineering, Massachusetts Institute of Technology, Cambridge, MA, USA. [6]Present address: Academy for Engineering and Technology, Fudan University, Shanghai 200433, China. [7]Present address: SanaHeal Inc., Cambridge, MA, USA. [8]These authors contributed equally: Camilo Duque Londono, Seth F. Cones, Jue Deng. ✉e-mail: amooney@whoi.edu; zhaox@mit.edu

challenges, and even lead to diseases and mortality in some soft-bodied animals[12]. These challenges hinder the wider use of electronic sensors to study social behaviors, distributions, and interactions of soft, fragile marine species.

Here, we report a bioadhesive interface for marine sensors (BIMS) to achieve rapid non-invasive integration (Fig. 1b) on soft fragile marine species including mollusca (squid, *Doryteuthis pealeii*) and cnidaria (jellyfish, *Aurelia aurita*). The BIMS contains a thin layer of a dried-hydrogel adhesive interface that can absorb the seawater from the surface of marine species by hydration and swelling, subsequently adhering to sensors on diverse marine species within 22 s. After adhering to the surface of marine species, the adhesive interface of the BIMS becomes a soft yet robust hydrogel. Owing to its fast non-invasive integration and matched mechanical properties with marine species, the BIMS does not disturb normal animal behaviors upon release and enables a wide suite of eco-physiological measurements and comparisons of the animals. We further demonstrate the BIMS

applicability to phylogenetically and morphologically distinct species, including actinopterygii (ray-finned fishes, *Salmo salar*, *Paralichthys dentatus*), chondrichthyes (skate, *Leucoraja erinacea*), and crustacea (lobster, *Homarus americanus*). The rapid adhesion across species further enables multi-joint and multi-animal sensor deployments to study in situ group behaviors, motion dynamics, and interactions between species.

## Results

### Design and mechanism of the BIMS

A thin dried-hydrogel adhesive interface (150 μm thickness) on the bottom of the BIMS enabled the rapid and non-invasive attachment of biologging sensors with marine species. This interface consisted of two interpenetrating polymer networks, which are physically crosslinked poly (vinyl alcohol) (PVA) and covalently crosslinked poly (acrylic acid) grafted with *N*-hydroxysuccinimide ester (PAA-NHS ester) (Fig. 1c). The adhesive interface in the dry state quickly absorbs interfacial water

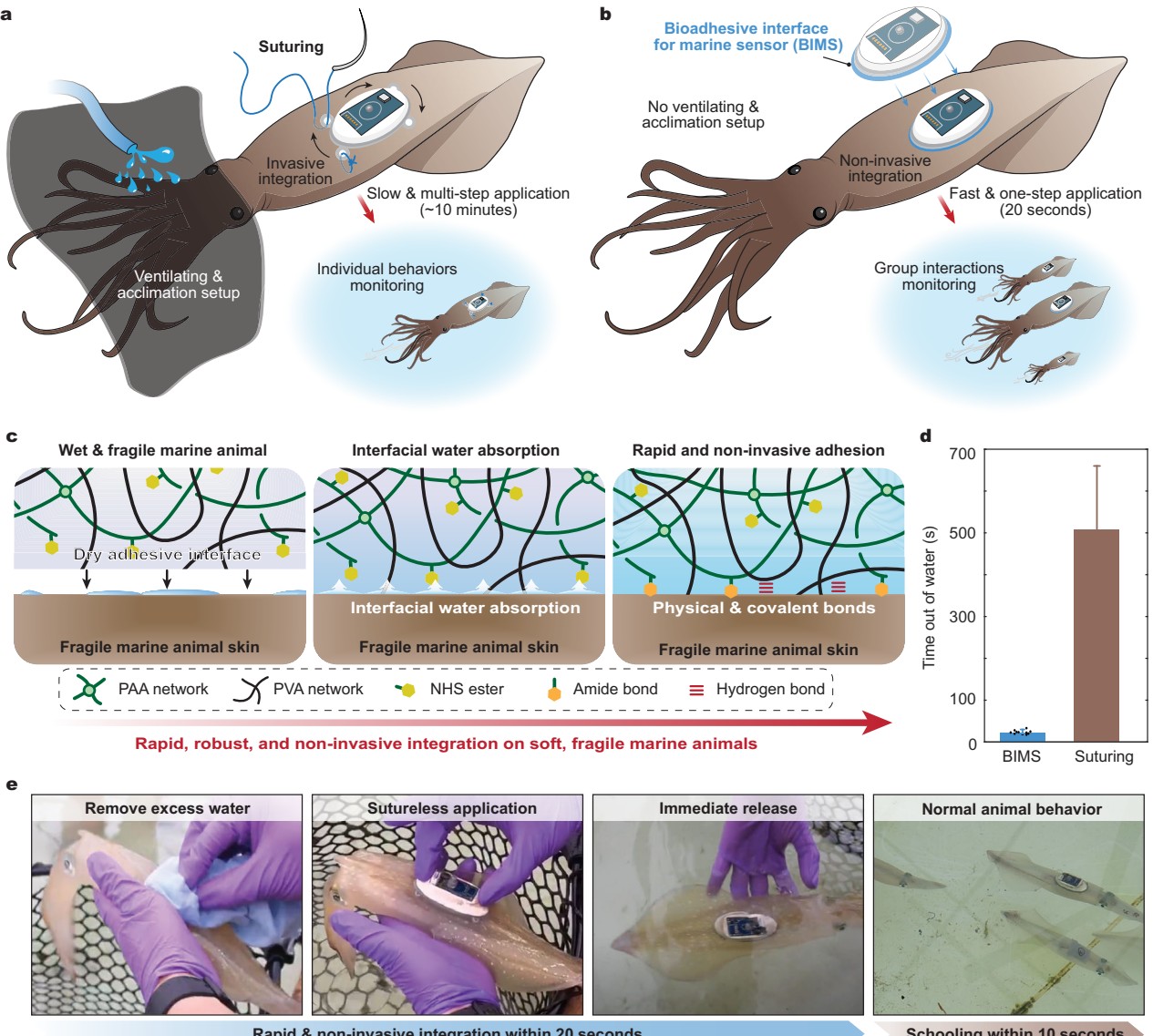

**Fig. 1 | BIMS mechanism and application process. a** Schematic illustration of surgical suturing sensors on the surface of a squid for monitoring individual animal behaviors. **b** Schematic illustration of rapid BIMS application on the surface of squid for group interactions monitoring. **c** Hydrogel adhesion mechanism for quick interfacial water absorption followed by physical and covalent crosslinks formation

with functional groups on the surface of marine animals. **d** Time out of the water for the BIMS and average time out of the water for surgical suturing (suturing application time data provided by Mooney et al. [19]). **e** Representative images of the BIMS application on a live squid. Values in (**d**) represent the mean and the range (*n* = 10 independent samples).

upon contact with marine species due to its hydrophilicity and hygroscopicity[13–15]. Carboxylic acid groups in the PAA-NHS ester form rapid adhesion through physical crosslinks (e.g., such as hydrogen bonds and electrostatic interactions), and subsequent covalent amide bonds between NHS-ester groups and primary amine groups on the marine animal tissue[16] ensured stable adhesion over time. The swollen adhesive interface also showed softness (Young's modulus of 250 kPa), high stretchability (over 5), and mechanical robustness (fracture toughness of over 1100 J m$^{-2}$), allowing conformable and imperceptible adhesion on curved tissue surfaces (Supplementary Fig. 1). Through surface chemical treatments, the adhesive interface can be easily coupled to a variety of soft engineering solids (i.e., silicone and urethanes), which can be used as an intermediate layer or an encapsulant for embedding sensors regardless of their material composition[17,18] (Supplementary Fig. 2).

To demonstrate the simplicity and quickness of the BIMS application compared to the current methods, we first used squid as a representative model to quantitatively evaluate the total time spent out of the water for tagging animals. The entire application process of the BIMS required an average of 22.3 s ($n = 10$), which is substantially faster than surgical suturing (8.5 min on average) based on the

previously reported results[19] (Fig. 1d). Moreover, the simple and rapid application process improves animal welfare and reduces stress on both the operators and the tagged animals. For example, tagging squid via complex and time-consuming surgical suturing requires stress-reducing measures such as covering the eyes[20] and constant ventilation of seawater into the mantle cavity[21] (Fig. 1a). Marine animals can incur significant physiological and metabolic stress during the tagging process, and behavior disruption can persist for numerous hours or even lead to mortality[22]. In contrast, the BIMS eliminated the need for ventilation and mitigating measures for prolonged air exposure and allowed all tag procedures to be completed by one person (Fig. 1e and Supplementary Movie 1).

## Adhesion performance

To evaluate adhesion performance of the BIMS across multiple marine animals, we first conducted three mechanical tests on ex vivo marine animal tissues, measuring the interfacial toughness, the shear strength, and the tensile strength (according to the following testing standards for tissue adhesives: ASTM F2256 for 90-degree peel tests, ASTM F2255 for lap-shear tests, and ASTM F2258 for tensile tests) (Fig. 2a–c and Supplementary Fig. 3). The adhesive interface of the BIMS formed a

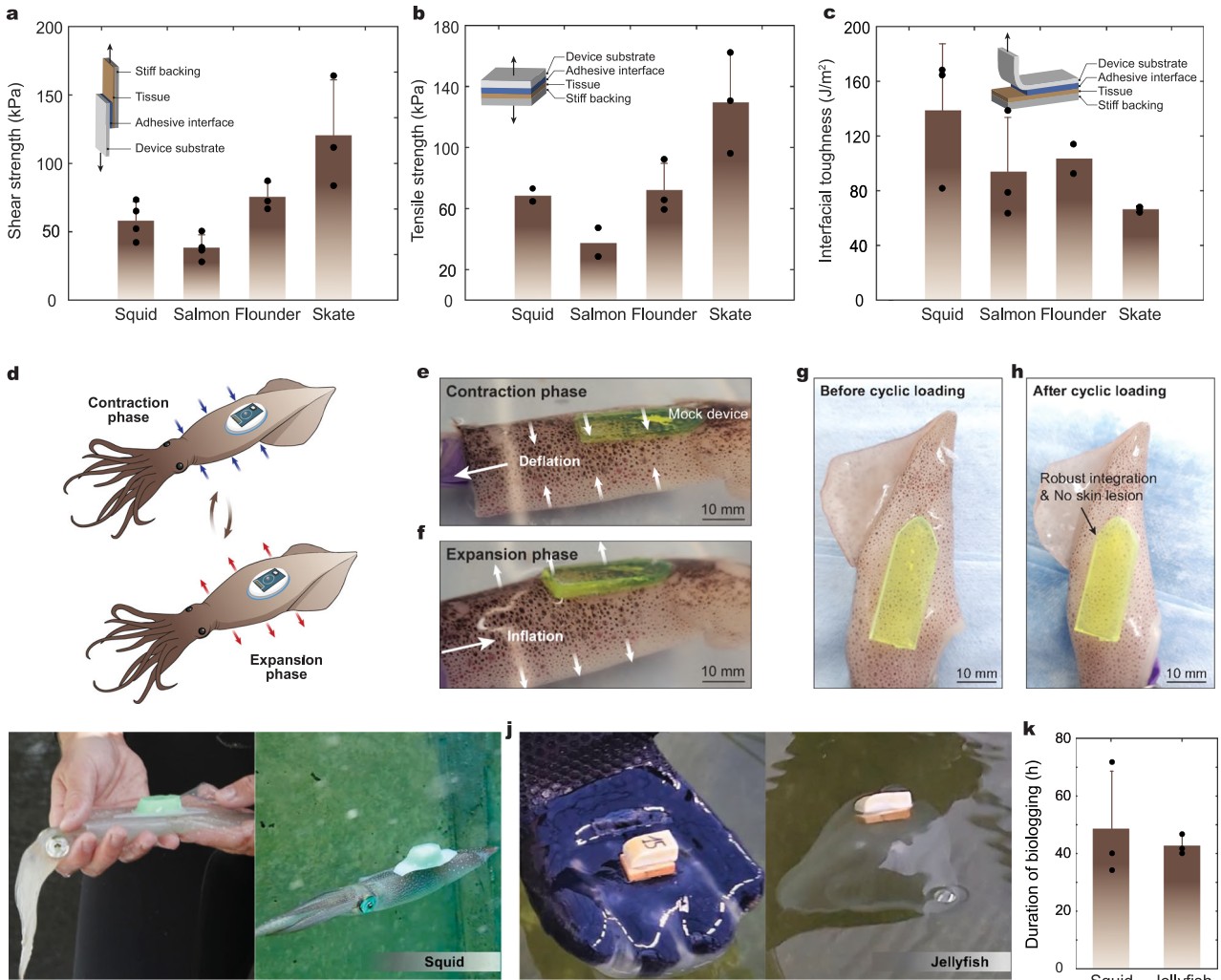

**Fig. 2 | Robust adhesion of the BIMS on diverse marine animals. a–c** Shear strength (**a**), tensile strength (**b**), and interfacial toughness (**c**) between the BIMS and various marine animal tissues. **d–f** Schematic illustration (**d**) and representative images (**e**, **f**) for a mock sensor adhered on a squid mantle with simulated respiration. **g**, **h** Representative images of robust and non-invasive adhesion on squid mantle before (**g**) and after (**h**) 40 cycles of simulated respiration. **i, j** Squid (**i**) and jellyfish (**j**) after tagging (left image in each panel) and freely swimming immediately upon release (right image of each panel). **k** Tag deployment time in water for squid and jellyfish. Values in (**a**, **b**, **c**, and **k**) represent the mean and the standard deviation ($n \geq 2$ independent samples).

rapid and robust bond to various wet tissues with a high interfacial toughness (>160 J m$^{-2}$ for squid, >130 J m$^{-2}$ for salmon, >70 J m$^{-2}$ for skate, and >70 J m$^{-2}$ for flounder), shear strength (>40 kPa for squid, >40 kPa for salmon, >80 kPa for skate, and >70 kPa for flounder), and tensile strength (>60 kPa for squid, >30 kPa for salmon, >130 kPa for skate, and >70 kPa for flounder). Adhesion stability and swelling behavior were further tested ex-vivo on squid tissue over a 24-hour period and at a wide range of temperatures found in the ocean (4 °C, 20 °C, and 30 °C). The lap shear strength was measured on squid tissue samples after 30 min, 1 h, and 24 h of adhesion. Shear strength on squid decreased an average of 15.6% in 24 h across the three temperatures. Notably, the shear strength was overall higher at 4 °C, likely due to the lower swelling ratio of 1.2 at 4 °C (Supplementary Fig. 4). Note that shear and tensile strength testing led to tissue failures instead of interfacial failures for squid, flounder, and salmon, shown by the detachment of the skin or scales (Supplementary Fig. 5).

For soft and fragile species, tagged sensors should conform with the organism's body shape. Rigid non-conformal sensors may induce epidermal lesions, which may further cause unmitigated bleeding, bacterial infection, and rapid mortality[12]. In addition, epidermal lesions could lead to poor data quality due to partially or fully displaced sensors. To evaluate adhesion robustness and non-invasiveness under cyclic body deformations, we applied the BIMS on fresh squid cadavers, and simulated animal respiration and movement by expanding and contracting the mantle cavity via pressurized air inputs in seawater. After 40 cycles of loading and unloading (Fig. 2d–h), the dermal layer around the BIMS application site remained intact with no visible tears or deformations. The elasticity of the swollen adhesive interface provided a medium for energy dissipation between the fragile skin and the rigid sensor as the skin stretches and contracts. Without this energy dissipation, the area immediately around the application site may suffer from stress or fracture, as may be the case with rigid glues such as cyanoacrylates. Therefore, the BIMS minimized potential tearing of surrounding tissue on animals that undergo body deformations while swimming such as squid, which expand their mantle cavity by roughly 41% when jetting[23].

## Rapid tagging and post-release behavior of live marine species

To assess the potential behavioral disruption by the BIMS, we measured the time taken for tagged squid to return to normal schooling behaviors with untagged animals (i.e., synchronized, directionally polarized groups). Upon release, squid were active and quickly swam towards untagged squid and began schooling (<20 s, $n = 5$) (Fig. 1e), demonstrating a rapid return to baseline individual and group-level behaviors. Natural, high-speed jet propulsion movements did not lead to adhesion failure, verifying strong resistance to dynamic loads while accelerating (Supplementary Movie 2). Importantly, untagged squid readily synchronized and schooled with the tagged squid without aggressive behaviors aimed toward focal squid, demonstrating that group-level behaviors were not impacted by the BIMS. In contrast, squid tagged via suturing typically need to recover in hyper-aerated coolers for fifteen minutes prior to reintroduction to conspecifics[19]. Without acclimation in coolers, squid are often lethargic and rest on the bottom for several minutes, which may increase individuals' susceptibility to predation in the wild.

Tag attachment mechanisms and methods are often species-specific. Such specialization has traditionally hampered method or tool scalability, broad applications to monitor multiple species concurrently, and greatly increases the training time and cost required to affix sensors. In contrast with existing species-specific methods, the simple and rapid features of the BIMS enabled instant tagging across multiple live marine species, including squid, jellyfish, skate, lobster, black sea bass (Centropristis striata), sea robin (Prionotus evolans), and flounder (Fig. 2i, j and Supplementary Fig. 6). We further evaluated the stability of the BIMS in a large tank (8 m in length, 2 m in width, and 0.75 m in depth). Due to the soft and robust characteristics of the adhesive interface, the BIMS can be reliably integrated with fragile aquatic animals, such as squid and jellyfish, for up to 3 days, allowing long-term animal behavior monitoring (Fig. 2k). All tags on live animals were left to fall off on their own, no tags were forcibly removed from live animals. After sensor detachment, individuals exhibited natural behaviors (i.e., schooling) and survived just as long as untagged animals in the experimental environment.

To highlight the rapid and reliable deployment of the BIMS, multiple sensors were adhered across a single animal. Four sensors (L, C, R1, R2) (Supplementary Fig. 7) were placed 100 mm from the anterior tip of an adult male skate. Sensors were spaced by 15 mm from L to R2, where L was placed on the left pectoral fin, C was placed at the centerline, and R1 and R2 divide the right pectoral fin. Surge acceleration for each point was directly measured while the skate swam in a flow tank (1.7 m in length, 0.5 m in width, and 0.6 m in depth) at a constant rate of 20 cm s$^{-1}$. As expected, due to symmetry, the acceleration signal from L and R1 are nearly identical, however, acceleration amplitude increased at R2. To expand on this, acceleration amplitude was plotted as a function of distance from the center (Supplementary Fig. 7) and fit with an exponential regression to map acceleration across the wing. While developing a full kinematic model is beyond the scope of this work, we show that the BIMS could aid kinematic studies, which currently rely on mathematical models, expensive computational models, and video motion analysis[24–26].

## BIMS functionality across species

To evaluate the robustness of the BIMS across diverse marine species and study the individual and inter-specific interactions via a series of sensors, we conducted a large-scale multi-species biologging experiment in a large 318,000-liter seawater aquaria (9 m in diameter, 5 m in height). The consistent application technique regardless of species facilitated tagging of a total of ten animals of four species within 25 min. At the individual level, we monitored ecologically important movement behaviors of squid (Fig. 3a, b), jellyfish (Fig. 3c, d), skate (Fig. 3e, f) and kelp (Supplementary Fig. 8) through three-dimensional acceleration and orientation via the BIMS. The individual data allowed us to identify the representative movement behaviors of specific animals[21,27], such as bi-directional swimming in squid, jet propulsion in jellyfish (Fig. 3a–d), and finning rates for squid (1.75 Hz) and skate (2.54 Hz) (Fig. 3a, e). The close contact between sensors and marine animals ensured high data quality and allowed for extracting detailed local motion patterns from diverse animals and movements.

Intraspecific aggregations and coordinated movements are widely observed behaviors across many evolutionary lineages[28,29]. In many marine species, grouping decreases predation risk[30], enables more effective navigation[31], and lessens metabolic demands[32]. Yet, group-level behaviors are rarely measured in situ due to the difficulty of monitoring multiple animals concurrently. Without any behavior disruption of either the focal or untagged animals within the groups for different marine animals (Fig. 4b–d), the BIMS played a pivotal role in collecting schooling data using an animal-borne sensor. The enhanced resolution in the BIMS relative to video analysis revealed the dynamics of coordinated schooling behavior and its evolution through a disruption (Fig. 4e–g). Squid preferentially organized into a highly structured school and individuals oriented mantle-first into the current, leading to strong alignment in both heading and pitch (Supplementary Movie 3). While squid schooled, an interspecific interaction was recorded via the BIMS, leading to decreased coordination demonstrated through increased orientation variance (Fig. 4g).

Community-level assessments of movement are poorly understood yet vital to measure energy flow and interactions within ocean habitats. To quantify and compare species movements (Fig. 4h), we calculated overall dynamic body acceleration (ODBA), which is an effective proxy for activity levels and energy expenditure (Halsey et al.,

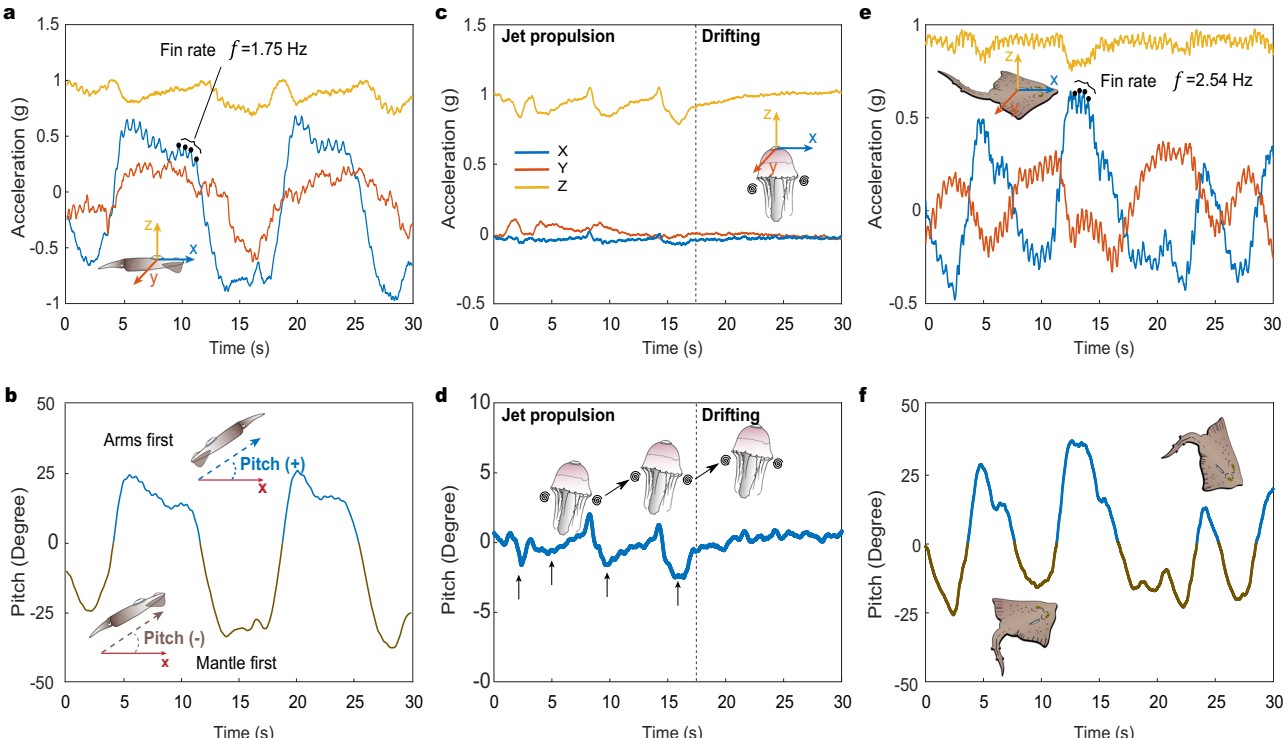

**Fig. 3 | Individual-level movement behaviors of diverse marine species measured by the BIMS. a**, **b** Acceleration (**a**) and pitch (**b**) of squid via the BIMS, showing the dominant frequency of fin rate (1.75 Hz) and the arms-first and mantle-first swimming orientation. **c**, **d** Monitoring of acceleration (**c**) and (**d**) of jellyfish via the BIMS, showing the jet propulsion events and passive resting behavior. **e**, **f** Real-time monitoring of acceleration (**e**) and (**f**) of skate via the BIMS, showing the dominant frequency of fin rate (2.54 Hz). Colors in (**a**, **c**, and **e**) represent direction, where blue is forward motion (surge, x), orange is lateral motion (sway, y) and yellow is vertical motion (heave, z).

2009) (Fig. 4g). The BIMS reliability across species facilitated concurrent assessment of movement strategies across multiple (>3) marine species and trophic levels. We found that ODBA levels showed a clear distinction between species with varying movement strategies. Flounder, a sit-and-wait ambush hunter, was largely sedentary and had the lowest average ODBA (0.032 g). The skate and lobster are more mobile benthic species, and their ODBA levels were greater and similar at 0.049 g and 0.052 g, respectively. Lastly, squid, which are muscular active pelagic swimmers, had the highest ODBA values (0.067 g). Such concurrent movement data across species could be calibrated and serve to measure energy flow, biotic interactions, and environmental constraints in dynamic marine communities, particularly in the context of the immense anthropogenic pressures arising from our changing oceans.

## BIMS in the field

To further evaluate the efficacy of the BIMS in the field, an ITAG[19] sensor was used with the BIMS, which was adhered on a free-ranging veined squid (*Loligo forbesii*) in the Azores Islands, Portugal (Fig. 5). Similar to our lab tests, we first removed excess interfacial water using a soft cloth (Fig. 5a). Next, the ITAG was gently pressed onto the squid mantle near the posterior mantle tip. Lastly, we took morphometric measurements and placed the animal immediately back into the water. The entire process was completed within 90 seconds (Supplementary Movie 4). The tagging procedure manifested a 466% enhancement in efficiency compared to traditional suturing methods[19]. For this field experiment, the adhesive was prepared in the lab and hand-carried to the Azores in bags filled with desiccant. The adhesive was kept protected from water and as dry as possible until use.

Upon release, the squid actively descended to a depth of 288 meters at an average rate of 35 cm s⁻¹ (Fig. 5b). During its descent, it sampled vertical oceanographic gradients, including temperature (Fig. 5c) and luminance of different light wavelengths (Fig. 5d). In addition, inertial measurement unit (IMU) sensors measured the movement dynamics and swimming behaviors while the squid was migrating downward. The squid showed natural swimming behaviors, alternating between periods of low (Fig. 5e, f) and high (Fig. 5g, h) intensity jet propulsion movements. Notably, the high intensity jet propulsion for this species averages 0.70 m/s[21] highlighting the robust adhesion capabilities of the BIMS, which can withstand drag induced by mobile species. The tag was released from the animal via a galvanic release system, effectively separating it from its base. Subsequently, it was tracked and recovered through a radio signal output. In this pilot field study, the length of deployment was limited by the timed release mechanism. Notably, the sensor retrieval without its base suggests that the base likely remained attached to the squid via the BIMS at the time of tag release. Based on observations from previous seawater tank experiments, we anticipate the BIMS to remain attached to a squid for a period ranging from 1 to 3 days before detaching on its own. This preliminary trial demonstrated that the BIMS can be suturelessly attached to monitor highly mobile free-ranging species that undergo rapid pressure changes in the field.

## Outlook

Despite the impressive miniaturization of marine sensors, flexible attachment technologies have proven challenging[4]. This study reports a universal platform to rapidly and non-invasively affix sensors on live marine species for studying individual, group, and multi-species behaviors beyond existing tag attachment mechanisms. We demonstrated its efficacy in a large-scale multi-species biologging experiment. The rapid and robust adhesive capability of the BIMS minimized the tagging effort and the need for training in species-

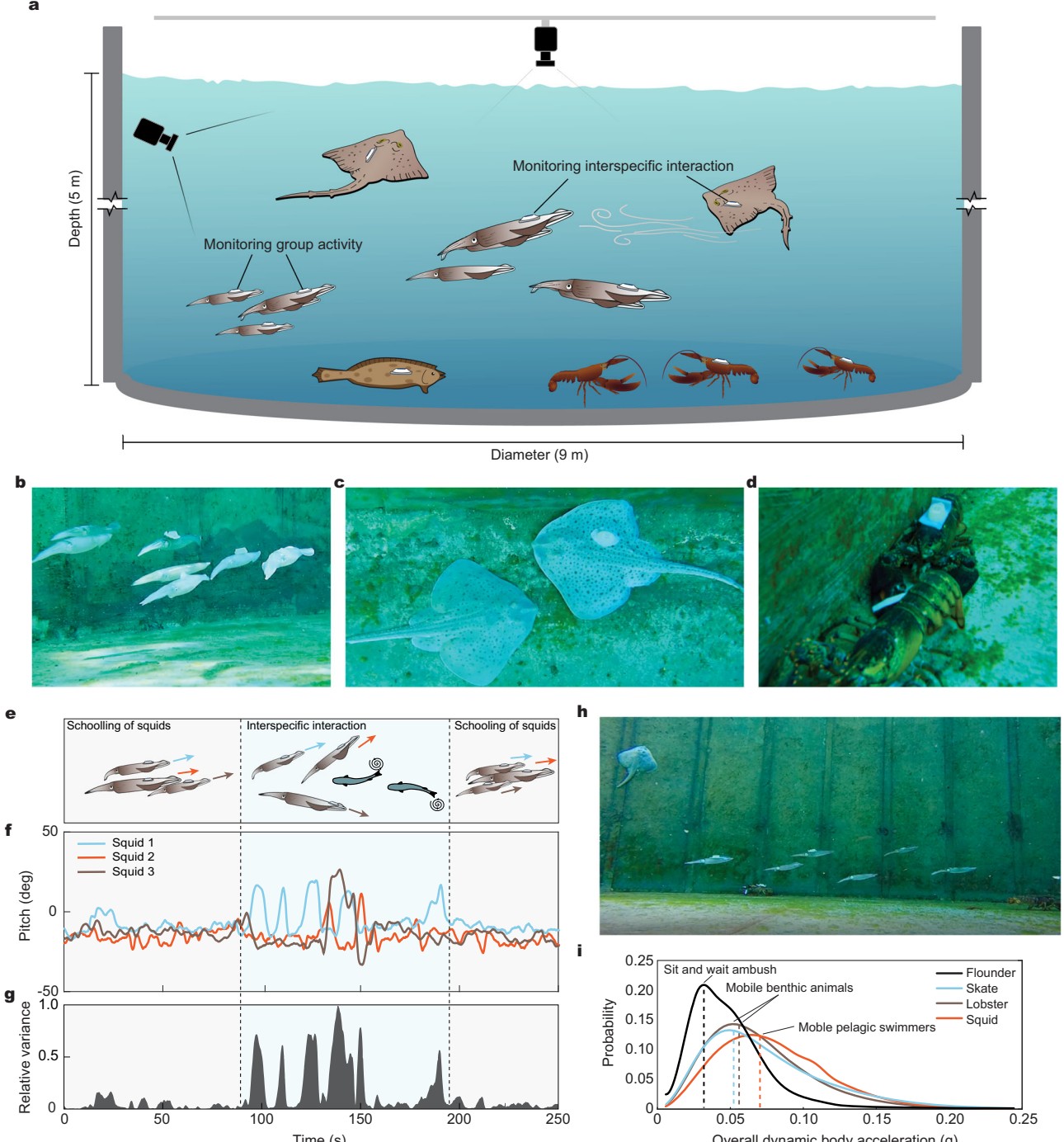

**Fig. 4 | Group and community behavior for marine animals equipped with the BIMS. a** Schematic of large-scale multi-species biologging experiment. Skate, squid, lobster, and flounder in a large saltwater tank (318,100 liters), half of the individuals per species were equipped with the BIMS. All animals were free-ranging and free-interacting within the tank. **b–d** Representative images of squid (**b**), skate (**c**), and lobster (**d**) equipped with the BIMS interacting with untagged animals. **e–g** Schematic (**e**), pitch (**f**), and relative variance (**g**) for three squid equipped with the BIMS interacting to an oncoming skate. Pitch and relative variance show group-level schooling behavior. **h** Representative image of interspecies interaction. **i** Average overall dynamic body acceleration (ODBA) of the flounder, skate, lobster, and squid monitored by the BIMS.

specific tagging methods. The non-invasive integration with marine species avoided behavioral disruption for either the focal or untagged animals, facilitating the collection of high-quality data. The approach also offers a promising avenue to expand the burgeoning field of marine animal sensing to small and soft-bodied invertebrates (e.g., squid, jellyfish). While we demonstrated that the BIMS is an effective tool for monitoring diverse marine species, future visions

can integrate more sophisticated acoustic or optical sensors[33]. Additionally, the BIMS may function as a drug delivery or wound protection device[34,35] for real-time therapies of injured or sick marine animals. Lastly, the BIMS may create sensor networks throughout the body of a single marine animal for locomotion kinematic studies, which could provide insight to improve the design and control of bio-inspired marine robots[36–40].

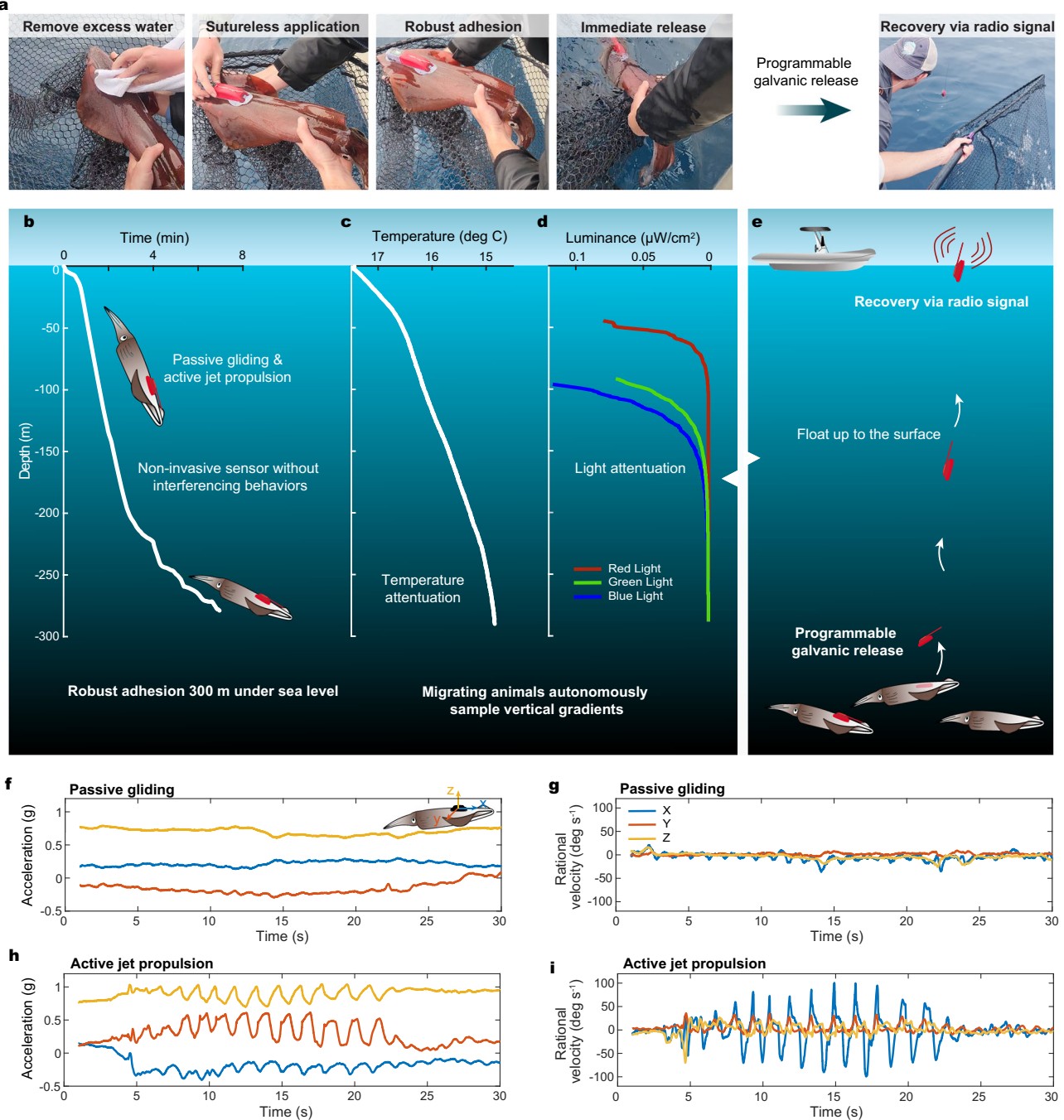

**Fig. 5 | BIMS application in the field. a** Representative images of the BIMS application in field, showing (left to right) drying, application, maneuvering, releasing, and tag recovery via radio signal. **b** The depth of the squid actively descends immediately after release as a function of time. **c, d** Adhered environmental sensors sampled the temperature (**c**) and luminance of RGB light (**d**) as a function of depth (**b**). **e** Schematic illustration of tag recovery via programable galvanic release and radio signal. **f–i** Accelerometer and gyroscope data for two ecologically important movement behaviors captured by the IMU. Gliding (**g**, **f**) represents a low-activity movement where the negatively buoyant animal descends in the water column. Jet propulsion (**h**, **i**) represents a high-activity movement and is often employed in series. Colors in (**f–i**), represent direction, where blue is forward motion (surge, x), orange is lateral motion (sway, y), and yellow is vertical motion (heave, z).

## Methods

### Materials

All chemicals were purchased from Sigma-Aldrich and used without further purification unless otherwise mentioned. To synthesize the hydrogel adhesive, acrylic acid (AAc), PVA (Mw = 146,000 to 186,000, 99 + % hydrolyzed), α-ketoglutaric acid, $N,N'$-bis(acryloyl)cystamine and $N$-acryloxysuccinimide (AAc-NHS) (VWR) were used. To prepare the silicone-encapsulated sensor device, Ecoflex 020 (Smooth-On) and

benzophenone were used. To attach the adhesive to the 3D-printed urethane base, hydrophilic polyurethane (PU) (HydroMed D3, Advansource Biomaterials) was used.

### Preparation of the adhesive interface

To prepare the stock solution for the adhesive, 35 w/w% AAc, 7 w/w% PVA, 0.2 w/w% α-ketoglutaric acid, and 0.05 w/w% $N,N'$-bis(acryloyl) cystamine was added to a bottle of nitrogen purged deionized water

equipped with a stirrer. The stock solution was stirred in a 90 °C water bath until completely dissolved. Once dissolved, this stock solution may be stored in a cool dry place in a UV-resistant glass bottle until use. To polymerize the stock solution and bolster its adhesive properties, 30 mg of AAc-NHS was dissolved for every 1 ml of the stock solution and then degassed to remove air bubbles. To make a thin, tape-like adhesive, two rectangular glass plates were treated with a hydrophobic coating (Rain-X) to allow for easier removal, and equipped with 150 μm spacers. The hydrogel precursor was then poured onto the glass mold between the spacers, clamped to form a seal, and cured in a UV chamber (365 nm, 10 W power) for 30 minutes. Care was taken to properly degass and seal the sample while in the UV chamber, failure to do this could inhibit polymerization. To reduce swelling during use, the hydrogel adhesive was stretched to a length and width equal to its equilibrium swelling ratio at room temperature of approximately 1.5, as shown in Supplementary Fig. 4, held with clamps and dried under continuous air flow for at least 3 h. While pre-stretching is not a requirement for the adhesive to work on tissues, it expedites drying and results in a more pliable film. The dried adhesive was then sealed in a bag with desiccant and stored at −20 °C until used.

## Coupling with electronic sensor

The silicone encapsulated sensor device was assembled from two separate parts. First, Ecoflex 020 was mixed for 3 min, degassed for 1 min, and poured onto a rectangular acrylic mold (150 × 180 × 2 mm) and allowed to cure at room temperature for 4 h. Curing in an oven under some heat will expedite curing time. Prior to curing, a flat piece of acrylic was used to scrape any excess silicone and ensure a flat sheet. To achieve a strong bond between the hydrogel adhesive and silicone, the cured silicone sheet was soaked in a solution of 10 w/v% benzophenone in ethanol for 10 min, rinsed with ethanol, and then air dried. The hydrogel adhesive precursor, prepared using the method above, was then poured onto one side of the benzophenone-treated silicone, sandwiched with a glass plate using 150 μm spacers, clamped and cured in a UV chamber for 30 min. Once cured, the silicone sheet with adhesive was air-dried for at least 3 h. For the sensor housing, Ecoflex 020 was prepared and poured on to a 3D printed mold with enough hollow space in the middle to tightly grip the sensor and cured for 4 h at room temperature. To finalize the silicone encapsulated sensor device, the sensor was inserted into the top mold and joined to the flat sheet using Sil-Poxy (Smooth-on). The device was sealed in an air-tight container with desiccant and stored at −20 °C until used.

To couple the dry adhesive layer to a 3D-printed urethane base, a thin layer of polyurethane was brushed onto the base. Then, the dry adhesive layer was pressed onto the wet polyurethane and allowed to dry under airflow for 10 min.

## Mechanical tests

All tissues were first soaked in artificial seawater, and prepared by dissolving 3.5 w/v% of sea salt (Sigma-Aldrich) in deionized water. All tissues were adhered to engineering solids using the hydrogel adhesive by pressing for 10 s. Mechanical tests were conducted at room temperature using a mechanical testing machine with a 2.5 kN load cell (Zwick/Roell Z2.5). A stiff nylon membrane, or thin polylactic acid film (Fischer Scientific) backing was applied using cyanoacrylate glue (Krazy Glue) to act as a stiff backing for the adhesive interface and marine animal tissues. Aluminum fixtures were applied using cyanoacrylate glues to provide grips for tensile tests.

To measure interfacial toughness, adhered samples with widths of 15 mm were tested by the standard 90-degree peel test (ASTM D2861). Peeling tests were conducted using a constant peeling speed of 50 mm min⁻¹. Interfacial toughness was determined by dividing the plateau force at a steady state by the adhesion width. Shear strength was measured using the standard lap shear strength test (ASTM F2255) using a constant tensile speed of 50 mm min⁻¹. Shear strength was

calculated by dividing the maximum force by the adhesion area, which was approximately 15 mm by 15 mm for each sample. Tensile strength was measured using a constant tensile speed of 50 mm min⁻¹ according to the standard tensile test (ASTM F2258). The tensile strength was calculated by dividing the maximum force by the adhesion area, which was approximately 15 mm by 15 mm for each sample.

Tensile properties and fracture toughness of the swollen adhesive interface were measured using pure-shear tensile tests of rectangular samples (30 mm in width, 10 mm in height, and 0.15 mm in thickness). Samples were equilibrated in seawater for 24 h prior to testing. Fracture toughness was calculated based on tensile tests of notched and unnotched samples of equal dimensions (Supplementary Fig. 1). All tests were done using a constant tensile speed of 50 mm min⁻¹ using a 20 N load cell (Zwick/Roell Z2.5). All data collected from mechanical tests was processed using Matlab 2023b.

## In vivo testing

To evaluate the efficacy of the BIMS across numerous species, we conducted a large-scale biologging experiment in a large (9 m diameter, 5 m height) saltwater pool at the Woods Hole Oceanographic Institution's (WHOI) Environmental Systems Laboratory (ESL). The experiment was conducted on October 19th, 2022 in a large saltwater tank filled with seawater (18 °C) from the nearby Vineyard Sound and was continually UV filtered. Animals were allowed to acclimate to the tank for 24 h prior to testing.

Two sensor packages were used in this study ITAGs contain a triaxial inertial measurement unit (IMU) with an accelerometer, gyroscope, and magnetometer sampled at 100 Hz (TDK Invensense MPU9250), and pressure, temperature (Keller 7LD), and light sensors (Intersil ISL29125) sampled at 1 Hz. The ITag was designed specifically for squid and jellyfish, with a hydrodynamic shape to limit additional lift and drag forces on the tagged animal, and was neutrally buoyant[19,41]. Secondly, Axy 5 XS bio-loggers (Technosmart Europe srl, www.technosmart.eu), measure tri-axial acceleration at 100 Hz and magnetic field strength in three axes at 2 Hz.

The tank was rigged with three GoPro Hero7 Black cameras for visual monitoring. All sensors and underwater cameras were time synchronized to accurately relate video and sensor data. Tags were affixed by removing the animal from the water, quickly patting the area of adhesion to remove any mucus and immediately applying the combined sensor and hydrogel adhesive with gentle pressure for 10 s. After, all animals were immediately released back into the water. The experimental animals were four lobsters (two tagged, total length: 25 cm, 22.5 cm), ten squid (five tagged, mantle length: 20.4 ± 2.6), three skate (two tagged, disc length: 37 cm, 42 cm), and one tagged flounder (total length 26 cm). Untagged and tagged animals were used to visually assess potential behavioral changes from applying the sensor with hydrogel adhesive and monitor interactions between tagged and untagged animals. Control animals were subject to the same tag process; they were removed from the water and handled as if a sensor was being applied, followed by immediate release. Animals for tagging were chosen at random, except for the flounder since only one was available. Sex was not considered in this study as a variable that would affect BIMS performance due to similarities in topological features across sex for the species tested. In addition, most species were not sexually dimorphic. Thus, identifying individual sex would require invasive probing and prolong tagging processes, which opposed a key goal of BIMS to rapidly and non-invasively adhere sensors.

## Movement and gait classification

Biologging tags were used to measure the acceleration and pitch of squid, jellyfish, skate, lobster, and flounder. For individual behaviors, raw acceleration (gravitational and dynamic) was reported in all three axes. We used a Fast Fourier Transform on the accelerometer axes to calculate finning rates for squid and skate[19]. Additionally, pitch was

calculated from the accelerometer (A) signal using $(A_x{}^2/(A_z{}^2 + A_y{}^2) - 0.5)$, and was used to measure swim direction of squid and detect water currents in kelp[19]. To compare movements across species, overall dynamic body acceleration was measured for all species (squid, skate, lobster, flounder, and kelp) in the large-scale marine sensing experiment. ODBA was used to compare movement rates among species and as a general activity metric. To distinguish between high-frequency animal movements (dynamic acceleration) and slower changes in orientation (gravitation acceleration), a low-pass filter (window size = 2.0 s[42]) was applied to all three axes of the accelerometer data. All data was analyzed using Matlab 2023b.

### Ethical statement and animal handling

This study was carried out in accordance with the principles of the Basel Declaration and recommendations and approval of the WHOI's Institutional Animal Care and Use Committee scientific protocol to TAM, IACUC number: 20811.03. Field testing in the Azores was completed with under the Azorean Regional Government's International Recognized Compliance Certificate 46/2021/DRCTD for access and use of natural resources for scientific purposes. All animal testing was done in compliance with local and national ethical regulations.

For tank experiments, skate, squid, sea robin, sea bass, and flounder were obtained from the Marine Resources Center at the Marine Biological Laboratory, Woods Hole, MA. Jellyfish were purchased from Sunset Marine Labs (Hillsboro, OR), transported overnight in water bags, and allowed to acclimate for 24 h. Live lobster was purchased from a local fish market. Skate, flounder, sea robin and sea bass were released into the nearby vineyard sound two weeks after testing. Local squid, jellyfish, and lobster were kept in their tanks indefinitely after testing. Squid used for field testing in the Azores was caught by jigging and immediately released upon tagging. Animal tissues for ex-vivo testing were purchased from a local fish market.

### Reporting summary

Further information on research design is available in the Nature Portfolio Reporting Summary linked to this article.

## Data availability

All data is available in the main text or supplementary information. Data in tabular form is available from the authors upon request.

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

## Acknowledgements

This work is supported in part by the Massachusetts Institute of Technology (X.Z.), the National Institutes of Health (Grants No. 1R01HL153857-01 and No. 1R01HL167947-01, X.Z.), the National Science Foundation (Grant No. EFMA-1935291, X.Z.; Grant No. 2203204, T.A.M.), and Department of Defense Congressionally Directed Medical Research Programs (Grant No. PR200524P1, X.Z.). C.D.L. acknowledges financial support from the Naval Sea Systems Command. In addition, this work was supported by the Bureau of Ocean Energy Management (M24AC00004-00, T.A.M). We thank Pedro Afonso and Jorge Fontes for their assistance testing BIMS in Faial, Azores Islands.

## Author contributions

S.F.C., C.D.L., H.Y., D.E.G., and X.Z. conceived the idea. S.F.C., C.D.L., J.D., and X.Z. designed the study. C.D.L., J.D., J.W., and H.Y. developed the materials and methods for the BIMS. C.D.L., and J.D. designed and performed the ex vivo experiments. S.F.C., C.D.L., and J.D. designed and performed the multi-species biologging experiments. S.F.C. and C.D.L. performed the field testing in the Azores. J.D., S.F.C., and C.D.L. analyzed and processed the data. T.A.M. and X.Z. supervised the study. S.F.C., J.D, C.D.L., T.A.M. and X.Z. wrote the manuscript with inputs from all authors.

## Competing interests

H.Y. and X.Z. are inventors of the bioadhesive technology used in the BIMS and have a financial interest in SanaHeal, Inc., a biotechnology company focused on the commercialization of the bioadhesive technology. The other authors declare no competing interests.
