## [Peer Review File · Nature Communications]

Bioadhesive Interface for Marine Sensors on Diverse Soft Fragile SpeciesREVIEWER COMMENTS

Reviewer #1 (Remarks to the Author):

In this manuscript, the authors developed bioadhesive marine sensor networks to study individual, group, and multi-species behaviors. The synergy of water absorption, physical crosslinks covalent amide bonds make the developed soft hydrogel-based bioadhesive achieving rapid, robust, and non-invasive integration on various marine animals. This work proposes a new axis of the application of soft hydrogel-based bioadhesive, and the reported results are interesting for the broad readers of Nature Communication, including biomaterials researchers, marine biologist, and engineer community. However, some revisions are needed to improve the integrity of the story considering the high impact of this journal.

1. The main concern is that the key component, namely soft hydrogel-based bioadhesive interface, some similar bioadhesives has been designed and fabricated in the previous study from the same authors (Nat. Mater. 20, 229–236 (2021); Sci Transl Med. 14(630), eabh2857 (2022)). Therefore, the novelty of material design is lacking.
2. The long-term stability of bioadhesive in various liquid environment need to be investigated, e.g., the temperature/salinity related swelling and adhesion properties of bioadhesive versus time.
3. In addition, the swelling behavior needs to be investigated.
4. Could BAMS be easily detached to avoid marine wear these sensors for life?
5. Is there any prospect of the developed BAMS being used in divers being used in divers or sailors?
6. Some demonstration experiments in real sea areas will further strengthen this manuscript.

Reviewer #2 (Remarks to the Author):

Species natural behaviour is very hard to study in the marine environment. The field relies on methods that document data under natural conditions when the human observer is absent. Tagging and biologging is a field in marine science that is rapidly advancing due to the development of ever smaller tags that can be used to track fish, sharks and cetaceans. However, a missing link in the field of biologging is the use of tags on soft bodied organisms such as molluscs and gelatinous organisms. These are abundant and ecologically pivotal taxa but their delicate bodies make it hard to attach tags to. Advancing our understanding of these bodied marine species relies on technological developments such as presented by Zhao et al. The authors present a new adhesive methodology which allows rapid and traumaless attachment of electronic tags on marine invertebrates. The authors describe the testing and adhesion procedure in great detail and even present very interesting preliminary data obtained by tagging various organisms in tank tests, to guide future research. I find the study very interesting, well prepared, well written, detailed and it surely breaks new ground in the field of biologging.

A major problem is that the authors do not explain how they obtain the tags or remove the tags from the animals to obtain the data. Do the animals survive after removal? Also I would like to have seen a discussion or an example set up how this new technology could be used in nature. Or if it is not possible to apply this technique in nature, the authors should explicitly state so.

In general, the authors do a great job listing the advantages of this approach, but they do not present a realistic view on the real application and its limitations. Therefore the manuscript is incomplete.

Reviewer #3 (Remarks to the Author):

Overall comments

For the marine biologging community this is a very important manuscript as it demonstrates a proof of concept for a new way forward for tagging soft fragile marine animals (invertebrates or but also smaller fragile vertebrates) which until now have largely been ignored because of the difficulty in attaching devices to their soft tissues. The authors demonstrate how a new type of bioadhesive can be used to attach tags to soft and fragile marine animals. The new adhesive is a hydrogel that is dried and then upon contact with the soft tissue of a marine animal absorbs water and starts to form a bond that attaches the tag to the animal. Very neat. This bond forms within 20 seconds or even less and so the animal can be placed back in the water and do not require ventilation or surgical procedures. The authors used 3 different mechanical tests to measure adhesive performance. The authors then demonstrate how multiple tags can be deployed on many individuals (across many species) so that schooling behaviour can be observed. The authors also discuss how multiple sensors can be deployed on a single individual to examine questions on ecophysiology. Overall it is an excellent manuscript with very important applications in the field of biologging. However, before it can be published I would like some structural changes/or concerns addressed and some further clarity on the writing of the manuscript. The authors should be congratulated on this work. I can't wait to use the adhesive!!!

Key comments: I find the use of term 'bioadhesive marine sensor networks' confusing throughout the manuscript. This stems from the fact that the authors have tagged both invertebrates and vertebrates but also a seaweed, so they need this wordy phrase to describe it. Messy. Why not drop the seaweed and perhaps just mention in a throwaway comment that adhesive also works on seaweeds etc.? Also, the most important finding from this study for me is the hydrogel and how easy it is to stick tags onto animals. This is publishable in itself with all the mechanical tests etc. and multiple species you tested it on. But this finding is muddled (in title and abstract) when the authors also discuss how multiple sensors can be deployed across many individuals and even species. So now the title becomes adhesive marine 'sensor network' (rather than "novel adhesive for attaching sensors/tags to soft and fragile marine animals") but I'm not sure how it is a network as individuals of a network communicate with each other and the sensors used in this study do not (I think but this is not clear to me)? Then finally the authors say they can attach multiple sensors to one animal. again this detracts from the main finding. I would prefer the authors just present findings on the adhesive and then say something like the speed (and non-invasiveness of procedure) at which tags can be deployed means that multiple tags can be deployed on

one animal (maybe show this in supplementary material). I would remove the sensors attached to many individuals/species showing schooling behaviour. This is a worthy finding of itself but needs further expansion and detail as hugely important work – how marine invertebrates aggregate/school, but it is lost in this new method manuscript.

See attached edited PDF with further comments.

Tom Doyle

ENDS

Response to Reviewer #1

Comment 1. In this manuscript, the authors developed bioadhesive marine sensor networks to study individual, group, and multi-species behaviors. The synergy of water absorption, physical crosslinks covalent amide bonds make the developed soft hydrogel-based bioadhesive achieving rapid, robust, and non-invasive integration on various marine animals. This work proposes a new axis of the application of soft hydrogel-based bioadhesive, and the reported results are interesting for the broad readers of Nature Communication, including biomaterials researchers, marine biologist, and engineer community. However, some revisions are needed to improve the integrity of the story considering the high impact of this journal.

Response 1. Thank you for your insightful and constructive comments. We greatly appreciate your recognition of the novelty and importance of our work. In the following paragraphs, we will address your comments point-by-point. The new sentences in the manuscript and supplementary information are marked in red.

Comment 2. The main concern is that the key component, namely soft hydrogel-based bioadhesive interface, some similar bioadhesives has been designed and fabricated in the previous study from the same authors (Nat. Mater. 20, 229–236 (2021); Sci Transl Med. 14(630), eabh2857 (2022)). Therefore, the novelty of material design is lacking.

Response 2. Thank you for the feedback. While the bioadhesive material compositions are adopted from the strategy previously developed in this lab group, we presented several improvements and insights in this work from material perspective. 1) We translate the adhesive hydrogel from biomedical applications to an entirely new field – marine animal biologging applications. The robust and ease of hydrogel adhesive technology is systematically evaluated in the marine environments, which have not been covered previously and will boost the applications of biologging. 2) We modified the adhesive hydrogel to be ready integrated with a variety of marine sensors, showing how this material could be widely integrated with existing systems.

Comment 3. The long-term stability of bioadhesive in various liquid environment need to be investigated, e.g., the temperature/salinity related swelling and adhesion properties of bioadhesive versus time. In addition, the swelling behavior needs to be investigated.

Response 3. Thank you for your insightful comments. As suggested, the swelling and adhesion were systematically investigated *ex vivo* over a 24-hour time period at three different temperatures (4°C, 20°C and 30°C). Considering the degradation of squid tissue soaked in water and the subsequent influence on the experiment results, the time of observation is no longer than 24 hours. Results show that the shear strength of the adhesive on squid tissue begins to decrease after 24 hours. Results on live animals also show that the adhesion to squid is stable for 24 hours. The adhesion strength is stronger overall at the lower temperatures due to the lower swelling ratio. The adhesion strength shows high stability over 24 hours at three different temperatures. We have added **Fig. R1** (added as **Supplementary Fig. 4**) and the following

paragraph in the revised manuscript and supplementary information.

“Adhesion stability and swelling behavior were further investigated ex vivo on squid over a 24-hour period and at three temperatures (4°C, 20°C, and 30°C). The lap shear strength was measured on squid samples after 30 minutes, 1 hour and 24 hours. Shear strength on squid decreased an average of 15.6% in 24 hours across the three temperatures. Notably, the shear strength was overall higher at 4°C, likely due to the lower swelling ratio of 1.2 (Supplementary Fig. 4).”

Fig. R1 | Swelling and adhesion stability at different temperatures. a,b, shear strength of the BAMS on squid tissue over 24 hours at three representative temperatures. c, Adhesive swelling ratio at three different temperatures. Values in a,b, and c are mean and standard deviation where error bars are present ($n = 3$ individual samples).

Comment 4. Could BAMS be easily detached to avoid marine wear these sensors for life?

Response 4. Based on our previous works (*Nat. Mater.* **20**, 229 (2021); *PNAS* **117**, 15497 (2020)), the covalent bonds and physical bonds (e.g., electrostatic interactions) in the adhesive interface can be easily cleaved via an aqueous biocompatible triggering solution (0.5 M sodium bicarbonate and 50 mM glutathione). While we did not include the benign detachment of the adhesive interface in this study, the sensors are able to detach from marine animals via galvanic release systems, leaving the adhesive base behind. In these set ups, an intermediary ‘base’ is placed between the sensors and the animal. Instead of sensors being directly attached to the animal, only the base is affixed. Then, a nichrome wire is used to connect sensors to the intermediary base. Because nichrome de-ionizes in seawater when an electrical current is applied, users can program specific ‘release times’ for on-demand detachment of sensors from the animals. The related schematic illustration and images have been added in the new Fig. 5.

Comment 5. Is there any prospect of the developed BAMS being used in divers or sailors?

Response 5. Thank you for your thoughtful feedbacks. As for now, BAMS cannot be employed underwater by divers or sailor. Nevertheless, the potential for underwater applications is definitely interesting. They could further decrease the out-of-water time for improved animal welfare and simplify the process of tagging marine animals that are challenging to capture from a boat. There is an ongoing project focused on the development of new BAMS for underwater applications, but this falls outside the scope of this work.

Comment 6. Some demonstration experiments in real sea areas will further strengthen this manuscript.

Response 6. BAMS was tested in real sea areas at the Azores, Portugal. In the **Fig. R2** (added as **Fig. 5** in the manuscript). we demonstrate that the squid was successfully tagged using BAMS in the field on the side of a boat and then released. The squid was observed diving to 300 meters under sea level with the sensor attached. Migrating squids autonomously sample vertical gradients of temperature and luminance.

Fig. R2 | BAMS application in the field. **a**, Representative images of the BAMS application in wild field, showing (left to right) drying, application, maneuvering, releasing, and tag recovery via radio signal. **b**, The depth of the squid actively descends immediately after release as a function of time. **c,d**, Adhered environmental sensors sampled the temperature (**c**) and luminance of RGB light (**d**) as a function of depth (**b**). **e**, Schematic illustration of tag recovery

via programmable galvanic release and radio signal. **f-i**, Accelerometer, and gyroscope data for two ecologically important movement behaviors captured by the IMU. Gliding (g and f) represents a low-activity movement where the negatively buoyant animal descends in the water column. Jet propulsion (h and i) represents a high-activity movement and is often employed in series. Colors in **f-i**, represent direction, where blue is forward motion (surge, or x), orange is lateral motion (sway, or y), and yellow is vertical motion (heave, or z).

Response to Reviewer #2:

Comment 1. Species natural behaviour is very hard to study in the marine environment. The field relies on methods that document data under natural conditions when the human observer is absent. Tagging and biologging is a field in marine science that is rapidly advancing due to the development of ever smaller tags that can be used to track fish, sharks and cetaceans. However, a missing link in the field of biologging is the use of tags on soft bodied organisms such as molluscs and gelatinous organisms. These are abundant and ecologically pivotal taxa but their delicate bodies make it hard to attach tags to. Advancing our understanding of these bodied marine species relies on technological developments such as presented by Zhao et al. The authors present a new adhesive methodology which allows rapid and traumaless attachment of electronic tags on marine invertebrates. The authors describe the testing and adhesion procedure in great detail and even present very interesting preliminary data obtained by tagging various organisms in tank tests, to guide future research. I find the study very interesting, well prepared, well written, detailed and it surely breaks new ground in the field of biologging.

Response 1. Thank you for the valuable feedback and interest in our project! We greatly appreciate your recognition of the novelty and importance of our work. In the following paragraphs, we will address your comments point-by-point. The new sentences in the manuscript and supplementary information are marked in red.

Comment 2. A major problem is that the authors do not explain how they obtain the tags or remove the tags from the animals to obtain the data. Do the animals survive after removal?

Response 2. BAMS was tested in real sea areas at the Azores. In the new Figure 5, we demonstrate that the squid was successfully tagged using BAMS and then released to sample vertical gradients of temperature and luminance autonomously. The sensors with obtained data are able to detach from marine animals via galvanic release systems. Specifically, the field tags are attached to a base plate which remains on the animal¹. The sensor itself is set to release after a pre-determined time, at which point the sensor separates from the base plate and floats to the surface. When surfaced, the sensor emits frequent pings on a known frequency (VHF) which allows for surface tracking and retrieval.

Experiments in enclosed tanks are employed to evaluate the animal's survivability after tag removal. The animals do survive after BAMS detaches. For example, multiple skate and black sea bass were tagged with BAMS and remained healthy over 1 month after BAMS detachment. Squid were observed swimming normally a few days after BAMS detachment. However, it was not possible to monitor squids for longer time periods due to the inherent issues with squid husbandry and other complications.

Comment 3. Also I would like to have seen a discussion or an example set up how this new technology could be used in nature. Or if it is not possible to apply this technique it in nature, the authors should explicitly state so. In general, the authors do a great job listing the advantages of this approach, but they do not present a realistic view on the real application and

its limitations. Therefore the manuscript is incomplete.

Response 3. BAMS was tested in real sea areas at the Azores. In the **Fig. R3** (added as **Fig. 5** in the manuscript) and **Supplementary Video 4**, we demonstrate that the squid was successfully tagged using BAMS in the field on the side of a boat and then released. The squid was observed diving to 300 meters under sea level with the sensor attached. Migrating squids autonomously sample vertical gradients of temperature and luminance.

Fig. R3 | BAMS application in the field. **a**, Representative images of the BAMS application in wild field, showing (left to right) drying, application, maneuvering, releasing, and tag recovery via radio signal. **b**, The depth of the squid actively descends immediately after release as a function of time. **c,d**, Adhered environmental sensors sampled the temperature (**c**) and luminance of RGB light (**d**) as a function of depth (**b**). **e**, Schematic illustration of tag recovery via programmable galvanic release and radio signal. **f-i**, Accelerometer, and gyroscope data for

two ecologically important movement behaviors captured by the IMU. Gliding (g and f) represents a low-activity movement where the negatively buoyant animal descends in the water column. Jet propulsion (h and i) represents a high-activity movement and is often employed in series. Colors in **f-i**, represent direction, where blue is forward motion (surge, or x), orange is lateral motion (sway, or y), and yellow is vertical motion (heave, or z).

Response to Reviewer #3:

Comment 1. For the marine biologging community this is a very important manuscript as it demonstrates a proof of concept for a new way forward for tagging soft fragile marine animals (invertebrates or but also smaller fragile vertebrates) which until now have largely been ignored because of the difficulty in attaching devices to their soft tissues. The authors demonstrate how a new type of bioadhesive can be used to attach tags to soft and fragile marine animals. The new adhesive is a hydrogel that is dried and then upon contact with the soft tissue of a marine animal absorbs water and starts to form a bond that attaches the tag to the animal. Very neat. This bond forms within 20 seconds or even less and so the animal can be placed back in the water and do not require ventilation or surgical procedures. The authors used 3 different mechanical tests to measure adhesive performance. The authors then demonstrate how multiple tags can be deployed on many individuals (across many species) so that schooling behaviour can be observed. The authors also discuss how multiple sensors can be deployed on a single individual to examine questions on ecophysiology. Overall it is an excellent manuscript with very important applications in the field of biologging. However, before it can be published I would like some structural changes/or concerns addressed and some further clarity on the writing of the manuscript. The authors should be congratulated on this work. I can't wait to use the adhesive!!!

Response 1. Thank you for the valuable feedback and interest in our project! We greatly appreciate your recognition of the novelty and importance of our work. In the following paragraphs, we will address your comments point-by-point. The new sentences in the manuscript and supplementary information are marked in red.

Comment 2. I find the use of term 'bioadhesive marine sensor networks' confusing throughout the manuscript. This stems from the fact that the authors have tagged both invertebrates and vertebrates but also a seaweed, so they need this wordy phrase to describe it. Messy. Why not drop the seaweed and perhaps just mention in a throwaway comment that adhesive also works on seaweeds etc.?

Response 2. Thank you for the feedback. We replace the term "bioadhesive marine sensor networks" into "bioadhesive for marine sensors" to avoid confusion. We simplify the discussion about seaweed and put the figures into supplementary information as Supplementary Fig. 8.

Comment 3. Also, the most important finding from this study for me is the hydrogel and how easy it is to stick tags onto animals. This is publishable in itself with all the mechanical tests etc. and multiple species you tested it on. But this finding is muddled (in title and abstract) when the authors also discuss how multiple sensors can be deployed across many individuals and even species. So now the title becomes adhesive marine 'sensor network' (rather than "novel adhesive for attaching sensors/tags to soft and fragile marine animals") but I'm not sure how it is a network as individuals of a network communicate with each other and the sensors used in this study do not (I think but this is not clear to me)?

Response 3. Thank you for your constructive feedback. We revise the title from “Bioadhesive marine sensor networks on Diverse Soft Fragile Species” to “**Bioadhesive for Marine Sensors on Diverse Soft Fragile Species**”. The use of the word sensor network throughout this manuscript is in the context of having groups of sensors from different species collecting data that would be retrieved and analyzed afterwards to inform individual, group, and community interactions. Although we wanted to highlight how this bioadhesive could facilitate studying marine ecosystems in a “sensor network” style as a big picture, we agree that may take away from the main finding, and have adjusted overall language throughout the manuscript.

Comment 4. Then finally the authors say they can attach multiple sensors to one animal. again this detracts from the main finding. I would prefer the authors just present findings on the adhesive and then say something like the speed (and non-invasiveness of procedure) at which tags can be deployed means that multiple tags can be deployed on one animal (maybe show this in supplementary material). I would remove the sensors attached to many individuals/species showing schooling behaviour. This is a worthy finding of itself but needs further expansion and detail as hugely important work – how marine invertebrates aggregate/school, but it is lost in this new method manuscript.

Response 4. We agree that the key finding of this work is the robust and rapid adhesive for tagging on marine species. To highlight this, we have moved the figure of multiple sensors on one animal to supplementary information as Supplementary Fig. 7. In addition to the exhaustive lab mechanical tests, the employment on more than one species also highlights the robustness of BAMS in large seawater environments where animals ‘interact’ with conspecifics, other species, and currents. The attachment to many individuals in a short time period also served as a showcase for the rapidness of employment. Therefore, it is important to keep the Figure which shows individual and group level behaviors from experiments in a 318000 liter seawater pool. Moreover, it is important to show the efficacy of this technology in the field. So, we developed a new figure showing a BAMS field test. The data presented in the **Fig. R4** (added as **Fig. 5** in the manuscript) and **Supplementary Video 4** is a brief summary of the field application demonstrating the ease of use in the field.

Fig. R4 | BAMS application in the field. **a**, Representative images of the BAMS application in wild field, showing (left to right) drying, application, maneuvering, releasing, and tag recovery via radio signal. **b**, The depth of the squid actively descends immediately after release as a function of time. **c,d**, Adhered environmental sensors sampled the temperature (**c**) and luminance of RGB light (**d**) as a function of depth (**b**). **e**, Schematic illustration of tag recovery via programable galvanic release and radio signal. **f-i**, Accelerometer, and gyroscope data for two ecologically important movement behaviors captured by the IMU. Gliding (**g** and **f**) represents a low-activity movement where the negatively buoyant animal descends in the water column. Jet propulsion (**h** and **i**) represents a high-activity movement and is often employed in series. Colors in **f-i**, represent direction, where blue is forward motion (surge, or *x*), orange is lateral motion (sway, or *y*), and yellow is vertical motion (heave, or *z*).

Comments in the annotated manuscript

Comment 5. This title is overly complicated. better to just say - A hydrogel adhesive for attaching sensors on soft fragile marine species.

Response 5. We revise the title from “Bioadhesive marine sensor networks on Diverse Soft Fragile Species” to “**Bioadhesive Marine Sensors on Diverse Soft Fragile Species**”.

Comment 6. In the previous manuscript, “*These methods can suffer from severe limitations, particularly for adhering on soft fragile marine species such as squid and jellyfish...*”. not always severe. dont think you need to say severe as not always. just delete severe./ enough to say these methods can suffer from limitations such as...

Response 6. The word “severe” has been deleted.

Comment 7. In the previous manuscript, “*Reliable and rapid adhesion of deployments multi-animal sensor-network to study biomechanics...*” it is not exactly clear what this means. suggest you make this very clear in abstract and methods and manuscript as currently not clear. so 1) you have tested a new hydrogel for attaching tags to animals and 2) you have demonstrated that this adhesive can be used to attach multiple tags to many different species BUT to me it is not clear if these tags communicate or if it is from video analysis and analysis of tag data that you gain insights?

Response 7. We revise this sentence into “**Reliable and rapid adhesion enables large scale, multi-animal sensor deployments to study biomechanics...**” to highlight the key finding and avoid confusion.

Comment 8. In the previous manuscript “*However, these sizeable and rigid sensors are not applicable to small fragile organisms, which are highly diverse, abundant (> 38.3 teragrams of carbon) and pervasive in all ocean habitats, as well as provide a key protein source for humans worldwide and sequester greenhouse gasses into the deep ocean.*” not all are small. some are just soft or fragile and lack hard carapaces of tough robust skin. replace with ' not applicable to attach to more fragile marine organisms that may have soft integuments, and/or may be very small e.g. many invertebrates such as squid and jellyfish but also many smaller fish species such as xx and xx.

Response 8. As suggested, we revised the sentence to provide a more exact description. “**However, these sizeable and rigid sensors are not applicable to attach to more fragile marine organisms that may be small or have soft outer protective layers as is true for many invertebrates such as squid and jellyfish, but also many small fish. Soft-bodied marine organisms are highly diverse, abundant (> 38.3 teragrams of carbon) and pervasive in all ocean habitats, as well as provide a key protein source for humans worldwide² and sequester greenhouse gasses into the deep ocean.**”

Comment 9. In the previous manuscript, “*The BAMS contains a thin layer of a dried-hydrogel adhesive interface that can absorb the seawater from the surface of marine species by hydration and swelling, on diverse marine species within 22 s. After adhering to the surface of marine species, the adhesive interface of the BAMS becomes a soft yet robust hydrogel.*” keep language simple. so once the dried hydrogel adhesive absorbs water from the surface of the marine species it sticks to the animal? no need to say subsequently integrating sensors??

Response 9. As suggested, we simplify this sentence. “**The BAMS contains a thin layer of a dried-hydrogel adhesive interface that can absorb the seawater from the surface of marine species by hydration and swelling, subsequently adhering sensors on diverse marine species within 22 s**”

Comment 10. In the previous manuscript, “*A thin dried-hydrogel adhesive interface (150 μ m thickness) on the bottom of the BAMS enable the rapid and non-invasive integration of*

biologging sensors with marine species.” replace integration with 'attachment'?? integration means that the tags is now part of the animal???

Response 10. As suggested, we replace “integration” with “**attachment**” in the revised manuscript.

Comment 11. In the previous manuscript, “*The adhesive interface can be easily coupled to a variety of soft engineering solids (i.e., silicone and polyurethane) and embedded with different sensors to be a universal BAMS*” not fully clear how sensors are inserted into hydrogel? do you insert sensors in the same way you 'stick' to animal i.e. just wet the top and attach sensor to wet part of hydrogel???? again language here is over complicated. the adhesive can be easily 'attached' to a variety of different materials such as silicone and polyurethane and then embedded with different sensors?

Response 11. We revise the sentence to avoid confusion. “**Through surface chemical treatments, the adhesive interface can be easily coupled to a variety of soft engineering solids (i.e., silicone and urethanes), which can be used as an intermediate layer or an encapsulant for embedding sensors regardless of their material composition, to be a universal BAMS.**”

Comment 12. In the previous manuscript, “*The adhesive interface of the BAMS forms a rapid and robust bond to various wet tissues with a high interfacial toughness ($>160 \text{ J m}^{-2}$ for squid, $>130 \text{ J m}^{-2}$ for salmon, $>70 \text{ J m}^{-2}$ for skate, and $>70 \text{ J m}^{-2}$ for flounder), shear strength ($>40 \text{ kPa}$ for squid, $>40 \text{ kPa}$ for salmon, $>80 \text{ kPa}$ for skate, and $>70 \text{ kPa}$ for flounder), and tensile strength ($>60 \text{ kPa}$ for squid, $>30 \text{ kPa}$ for salmon, $>130 \text{ kPa}$ for skate, and $>70 \text{ kPa}$ for flounder).*”no mention of jellyfish here? yet mentioned in next sentence.

Response 12. Mechanical tests on jellyfish tissue were not possible due to difficulty reliably fixing the gelatinous underside of the tissue to the testing device. Jellyfish in the next sentence is deleted.

Comment 13. In the previous manuscript, “*In addition to facilitating multi-animal sensor network deployment, the BAMS allows for rapid and reliable adhesion of multiple sensors across a single animal.*” this is possible with many different tags so not sure what the message is here?

Response 13. We simplify this sentence. “**To highlight rapid and reliable deployment of BAMS, multiple sensors are adhered across a single animal.**” The figure associated to this paragraph is moved to Supplementary Information.

Comment 14. In the previous manuscript, “To evaluate the robustness of the BAMS across diverse marine species and study the individual, group, and community behaviors via a sensor network”. You do not explain what you mean by a sensor network? do the tags communicate with each other or simply is it lots of tags on many different animals in the same place?

Response 14. The tags do not communicate with each other in this work. To avoid any potential confusion, we have replaced the term “a sensor network” into “a series of sensors.”

Comment 15. What does Fig. 2k show? time tag remained on animal in water? not clear.

Response 15. Fig. 2k shows “the duration of biologging” on squid and jellyfish via BAMS. We revise the figure accordingly.

Comment 16. In Figure 4. need to explain different colours in caption too. not enough to have little figure of jellyfish showing x, y and z and spiral?

Response 16. The explanation of different color has been added in caption. “Colors in **a,c,e,g**, represent direction, where blue is forward motion (surge, or x), orange is lateral motion (sway, or y) and yellow is vertical motion (heave, or z).”

REVIEWER COMMENTS

Reviewer #1 (Remarks to the Author):

My concerns have been addressed by the authors. An acceptance for publication of this paper is thus recommended.

Reviewer #3 (Remarks to the Author):

I am very happy that the authors have addressed all my comments and I am happy for this to be published. I very much look forward to seeing this published.

Reviewer #4 (Remarks to the Author):

Bio adhesive review NCOMMS-23-19712A

I was asked by the editor to comment on the author's responses to Reviewer 2 to see if I thought they adequately dealt with the reviewer's concerns. The reviewer produced 3 comments. The first was a statement praising the work. It required no significant reply and the authors express thanks for the praise.

The second raised two questions: They pointed out that the authors did not explain how they "obtained the tag"s or removed the tags from the animals to recover the data. Im was not certain if the review want to know the source of the electronic s they used, i.e commercially sourced or specially custom built in house/ to on not certain but is seems that the answer might be some of both. . I am aware of several suppliers of similar recording instruments.

The authors responded to this comment by adding a significant new subsection to the paper that explains the field procedure in more detail and that included an informative short video of the animal being tagged and released. They explained that the tag (which record, rather than transmit, the data they collect) was "released from the animal via a galvanic release system, and that they "recovered the tag via a radio signal output". They also added a new figure that provides a pictorial schematic of the field process as well as the informative video. I expect that this might satisfy the reviewer but, if it were me, I would like to have a bit more detail, particularly on the detail of the preparation of the adhesive. The section of the paper captioned "Preparation of the adhesive interface" is key to enabling practitioners to use the attachment approach. It is brief and, as someone not schooled in such engineering, I would need some further discussion about the process, the rationale for the various steps and potential pitfalls and criticalities before I attempted to use it.

Reviewer 2 also asks, "Do the animals survive after removal". The authors suggest that they do, in the short term. There is no data on long term effects, if any. This would of course be very useful, if available.

One thing that was not made clear is whether, after release of the recording package, what remains on the animal. If so, how long will it remain attached and how might it affect the tissue during that time. I would have liked more detailed info on the removal of the adhesive as they attempted to show in Sup. Fig. 6. I did not find the images very helpful so perhaps they could provide some clearer images or some diagnostic discussion of any apparent injury that might have resulted from pressure, shear or necrotic tissue under the attachment.

The 3rd comment I guess was driven by a lack of any field study in the original manuscript so this whole new study and the associated text and figure should make it clear that the use of the attachment method is feasible in the field.

Some gratuitous comments from my own reading of the paper. For me, the key aspect of this paper is the report on a potentially very useful adhesive. I don't view that data collected on the animals as particularly useful. To me, its function in the paper is simply an example of how the adhesive might be used for attachment of instruments using the adhesive on animals with a variety of integuments. The adhesive seems similar to some that has been developed and used previously in surgical interventions on wet, soft tissues and that have advantages over conventional suturing and other tissue glues like cyanoacrylates e.g. DOI: 10.1126/science.abo2542; doi: 10.1016/j.bioactmat.2021.10.001; <https://sanaheal.com/> .

Here the authors show it is also suitable for attachment of instruments to soft bodied marine animals. The limitations of existing methods and the importance of finding better attachment methods is appreciated by the biologists studying such creatures and this is an important new application of the medical methodology in this new realm.

But the paper seems to conflate two objectives. One is the description of the adhesive, its preparation and mechanical testing via actual application to a set of animals in the lab where they can be observed for a time to assess the efficacy of the attachment; the other is to demonstrate that the methodology is applicable in the field by providing an example of how it might be used on a soft-bodied animal, a squid, with a particular data recording device that records data on depth, acceleration and light level. (Such devices are now often used and I believe are commercially available so this aspect is not novel.)

The title reads, "Bioadhesive Marine Sensors on Diverse Soft Fragile Species". They say, "Here we introduce a soft hydrogel-based bioadhesive marine sensor (BAMS)..." The original title probably obscured the contents even more in that a reviewer asked, "In this manuscript, the authors developed bioadhesive marine sensor networks [*italics mine*] to study individual, group, and multi-species behaviors." The importance of the paper is the adhesive. The sensor is independent of the glue. There is no network involved, let alone data on group behaviours. They seem to be suggesting some sort of tags that communicate with each other. It could be any sort of tag.

It seems to me that the authors are combining a number of different objectives in the title and the introduction in a way that I think will be confusing to the reader. What they are predominantly doing is calling attention to a type of surgical adhesive developed to join or repair soft tissues in a medical context that may also provide an effective way of attaching things to slippery, flexible, animals. Importantly, this adhesive is both stretchy and will bond to wet surfaces. This is a very important advance. Because the glue is stretchy, the animals don't suffer so much from the compliance mismatch between the stiff instruments being attached and therefore deforming the living tissue causing injury as that animals bend and move. This mismatch can also cause instruments to drop off sooner than they otherwise would.

This could be an important advance, well worthy of prominent publication. (They perhaps should note that

this compliance mismatch is also an important constraint in large, robust animals like dolphins and even in seals, where tags can be glued to hair. In such animals, a good compliant, fast setting glue would really be useful too. Big animals bend at scales where larger tags can pull away from the skin or fur causing them to come away prematurely. Parenthetically, there are people trying to actually produce flexible electronics that will themselves bend and stretch e.g. . “Flexible Bioelectronic Tag with a Kirigami-Based Design for Crosstalk Suppression in Multimodal Sensing”, Qiwei He, et al, <https://doi.org/10.1002/admt.202300982>)

But glue is not the “sensor”; nor is it a “network”. Why conflate these disparate things? This derails a potentially clear simple report of an important idea.

I think the paper would benefit from being reorganized on the basis of its 3 parts. They should just follow the thought process that must have gone on. Firstly, mention the surgical glue (and reference that original paper), then discuss the general characteristics that make it effective in soft tissue repair. Then they should note that these characteristics are useful for an application in an entirely different realm, i.e. animal biotelemetry, where its use for attachment to soft bodied animals will make possible studies that are presently very difficult or unfeasible. Then description of the glue and its characteristics including its compliance and the testing of mechanical properties should follow in turn followed by, a description of the laboratory tank testing on a variety of animals. This section should provide more information, including a clear description of the estimated how long the tags would stay on the animals. After all, this is the first question someone designing a sturdy using the glue would ask.

They say, “Due to the soft and robust characteristics of the adhesive interface, the “BAMS” can be reliably integrated with fragile aquatic animals, such as squid and jellyfish, for up to 3 days, allowing long-term animal behavior monitoring (Fig. 2k). Does this mean that 3 days is the maximum time the bond will last? How about the actual data on attachment time on the different animals. This is what practitioners will want to know, not the data on ODBA etc. In general, the data from the tags is not the important part of the paper, only serving as an example of potential uses. Anyway, it is not presented in a biologically meaningful way. It is just an anecdotal account of what tags might record if used with this attachment method. I guess reporting the ODBA results shows that the mounting is not so compliant that it damps out the acceleration signal to a degree that obscures indications of animal activity but that is not the point that the text makes. It suggests different animals wiggle in different ways, not much more than that.

Figures 3 and 4 serve to show how tags might be used. The authors say in Fig 4. , “Group and community behavior for marine animals equipped with the BAMS. a, Schematic illustration of large-scale multi-species biologging experiment. Skate, squid, lobster, and flounder were placed in a large saltwater tank (318,100 liters), and roughly half of the individuals of each species were equipped with the BAMS”.

Again, this is not really a biologging experiment to study animal behavior, rather it is an example of the sorts of animals on which that the attachment adhesive might be used. The value of the detailed data serve only that the attachment works. There is not natural community to study here.

And finally, Fig 5 provides a short description of the field test, and its value is primarily to show that the adhesive can be used in the field. It would have been useful to go into a bit more detail of exactly where, when and how the adhesive was prepared, stored and transported to where the animals were caught.

The details of the data obtained during this test is of little relevance to the main value of the paper, which is purely the development and testing of the glue. The tag could just as well been a mock up. A single depth profile of light level and associated acceleration is of little significance in its own right. Here,

the figure simply provides an example of how the attachment might be used.

There should be more discussion of the mechanics of the bond as it decays over time due to biodegradation. The most important thing missing for me was the detail on how long the tags would have stayed on if they were not actively removed and the state of the tissue after the attachment falls or is pulled off. In the tanks, were the tags actively removed or left on until they dropped off? I understand that there was an issue with maintenance of the squid in captivity, but how long were tags left attached? Were they actively removed? Were the mechanical properties of the glue changed by longer exposure to sea water? What did the scales or skin of the animals look like after forced or “eventual” detachment? These are the sort of details that I think the bio-logging community would like to hear more about so they could better understand what sorts of studies the glue would facilitate.

Comment 1. I was asked by the editor to comment on the author's responses to Reviewer 2 to see if I thought they adequately dealt with the reviewer's concerns. The reviewer produced 3 comments. The first was a statement praising the work. It required no significant reply and the authors express thanks for the praise.

***Comment 1 from initial Reviewer 2.** Species natural behaviour is very hard to study in the marine environment. The field relies on methods that document data under natural conditions when the human observer is absent. Tagging and biologging is a field in marine science that is rapidly advancing due to the development of ever smaller tags that can be used to track fish, sharks and cetaceans. However, a missing link in the field of biologging is the use of tags on soft bodied organisms such as molluscs and gelatinous organisms. These are abundant and ecologically pivotal taxa but their delicate bodies make it hard to attach tags to. Advancing our understanding of these bodied marine species relies on technological developments such as presented by Zhao et al. The authors present a new adhesive methodology which allows rapid and traumaless attachment of electronic tags on marine invertebrates. The authors describe the testing and adhesion procedure in great detail and even present very interesting preliminary data obtained by tagging various organisms in tank tests, to guide future research. I find the study very interesting, well prepared, well written, detailed and it surely breaks new ground in the field of biologging.*

Response 1. Thank you for your time to review our revision manuscript and supplementary information and confirming that we address **Comment 1 from initial reviewer 2**

Comment 2. The second raised two questions: They pointed out that the authors did not explain how they "obtained the tag"s or removed the tags from the animals to recover the data. Im was not certain if the review want to know the source of the electronic s they used, i.e commercially sourced or specially custom built in houseI to on not certain but is seems that the answer might be some of both. . I am aware of several suppliers of similar recording instruments.

The authors responded to this comment by adding a significant new subsection to the paper that explains the field procedure in more detail and that included an informative short video of the animal being tagged and released. They explained that the tag (which record, rather than transmit, the data they collect) was "released from the animal via a galvanic release system, and that they "recovered the tag via a radio signal output". They also added a new figure that provides a pictorial schematic of the field process as well as the informative video. I expect that this might satisfy the reviewer.

***Comment 2 from initial Reviewer 2.** A major problem is that the authors do not explain how they obtain the tags or remove the tags from the animals to obtain the data. Do the animals survive after removal?*

Response 2. Thank you so much to confirm our newly added experimental data and video, which might satisfy the **Comment 2 from the initial reviewer 2.**

Comment 3. but, if it were me, I would like to have a bit more detail, particularly on the detail of the preparation of the adhesive. The section of the paper captioned “Preparation of the adhesive interface” is key to enabling practitioners to use the attachment approach. It is brief and, as someone not schooled in such engineering, I would need some further discussion about the process, the rationale for the various steps and potential pitfalls and criticalities before I attempted to use it.

Response 3. The particular part of the methods section was enhanced with more detail on the preparation of the hydrogel adhesive. Explanations and potential pitfalls for some of the key steps were also included.

Comment 4. Reviewer 2 also asks, “Do the animals survive after removal?”. The authors suggest that they do, in the short term. There is no data on long term effects, if any. This would of course be very useful, if available. One thing that was not made clear is whether, after release of the recording package, what remains on the animal. If so, how long will it remain attached and how might it affect the tissue during that time. I would have liked more detailed info on the removal of the adhesive as they attempted to show in Sup. Fig. 6. I did not find the images very helpful so perhaps they could provide some clearer images or some diagnostic discussion of any apparent injury that might have resulted from pressure, shear or necrotic tissue under the attachment.

The 3rd comment I guess was driven by a lack of any field study in the original manuscript so this whole new study and the associated text and figure should make it clear that the use of the attachment method is feasible in the field.

Comment 3 from initial Reviewer 2. Also I would like to have seen a discussion or an example set up how this new technology could be used in nature. Or if it is not possible to apply this technique in nature, the authors should explicitly state so. In general, the authors do a great job listing the advantages of this approach, but they do not present a realistic view on the real application and its limitations. Therefore the manuscript is incomplete

Response 4. Thank you for confirming that the newly added field study makes the **Comment 3 from initial reviewer 2** clear.

In terms of the survivability after removal, marine animals survive and behave normally after detachment, and no adhesive remains on the animal after detachment. Tagged and tag-detached animals survived just as long as untagged animals, and showed no noticeable change in behaviors. Squid continued to school with untagged animals and fishes continued to swim, feed, and hide in shelters. Note that squid caught from sea and farmed in experimental environments typically survive for roughly 7 days, due to the inherent issues and complications (i.e., skin damage from swimming into tank walls). The attachment and detachment of the sensors via bioadhesive interface does not reduce squid survival time. The adhesive interface allows 3-day attachment on live squid. In our experiments, we did not attempt to remove the adhesive from a live animal, all samples were left to detach on their own. In the field study, the sensor detached from a base with a programmed time release. Although not possible to observe

in the wild, based on our evidence, the base would have remained on the animal until the adhesive failed, at which point the base would fall off the squid with the adhesive on it.

Comment 5. Some gratuitous comments from my own reading of the paper. For me, the key aspect of this paper is the report on a potentially very useful adhesive. I don't view that data collected on the animals as particularly useful. To me, Its function in the paper is simply an example of how the adhesive might be used for attachment of instruments using the adhesive on animals with a variety of integuments.

The adhesive seems similar to some that has been developed and used previously in surgical interventions on wet, soft tissues and that have advantages over conventional suturing and other tissue glues like cyanoacrylates e.g. DOI: 10.1126/science.abo2542; doi: 10.1016/j.bioactmat.2021.10.001; <https://sanaheal.com/> .

Here the authors show it is also suitable for attachment of instruments to soft bodied marine animals. The limitations of existing methods and the importance of finding better attachment methods is appreciated by the biologists studying such creatures and this is an important new application of the medical methodology in this new realm.

Response 5. Thank you for the feedback. While the bioadhesive material compositions are adopted from the strategy previously developed in this lab group, we presented several improvements and insights in this work from a material perspective. 1) We translate the adhesive hydrogel from biomedical applications to an entirely new field – marine animal biologging applications. The robust and ease of hydrogel adhesive technology is systematically evaluated in the marine environments, which have not been covered previously and will boost the applications of biologging. 2) We modified the adhesive hydrogel to be ready integrated with a variety of marine sensors, showing how this material could be widely integrated with existing systems.

As suggested, we revise the title and manuscript accordingly to highlight the novelty of bioadhesive interface instead of marine sensors. We also respectively point out that real-time sensing data collected from various marine species provide representative examples and more opportunities for future researchers in the field of biologging and beyond to 1) identify animal behaviors, 2) estimating orientation, and 3) collect environmental data to contextualize behaviors. Therefore, seamlessly and easily collecting various data from soft marine animals highlights the novelty and importance of this work.

Comment 6. But the paper seems to conflate two objectives. One is the description of the adhesive, its preparation and mechanical testing via actual application to a set of animals in the lab where they can be observed for a time to assess the efficacy of the attachment; the other is to demonstrate that the methodology is applicable in the field by providing an example of how it might be used on a soft-bodied animal, a squid, with a particular data recording device that records data on depth, acceleration and light level. (Such devices are now often used and I believe are commercially available so this aspect is not novel.)

The title reads, “Bioadhesive Marine Sensors on Diverse Soft Fragile Species”. They say, “Here we introduce a soft hydrogel-based bioadhesive marine sensor (BAMS)....” The original title probably obscured the contents even more in that a reviewer asked, “ In this manuscript, the

authors developed bioadhesive marine sensor networks [italics mine] to study individual, group, and multi-species behaviors.” The importance of the paper is the adhesive. The sensor is independent of the glue. There is no network involved, let alone data on group behaviours. They seem to be suggesting some sort of tags that communicate with each other. It could be any sort of tag.

It seems to me that the authors are combining a number of different objectives in the title and the introduction in a way that I think will be confusing to the reader. What they are predominantly doing is calling attention to a type of surgical adhesive developed to join or repair soft tissues in a medical context that may also provide an effective way of attaching things to slippery, flexible, animals. Importantly, this adhesive is both stretchy and will bond to wet surfaces. This is a very important advance. Because the glue is stretchy, the animals don't suffer so much from the compliance mismatch between the stiff instruments being attached and therefore deforming the living tissue causing injury as that animals bend and move. This mismatch can also cause instruments to drop off sooner than they otherwise would.

This could be an important advance, well worthy of prominent publication. (They perhaps should note that this compliance mismatch is also an important constraint in large, robust animals like dolphins and even in seals, where tags can be glued to hair. In such animals, a good compliant, fast setting glue would really be useful too. Big animals bend at scales where larger tags can pull away from the skin or fur causing them to come away prematurely. Parenthetically, there are people trying to actually produce flexible electronics that will themselves bend and stretch e.g. . “Flexible Bioelectronic Tag with a Kirigami-Based Design for Crosstalk Suppression in Multimodal Sensing”, Qipei He, et al, <https://doi.org/10.1002/admt.202300982>)

But glue is not the “sensor”; nor is it a “network”. Why conflate these disparate things? This derails a potentially clear simple report of an important idea.

Response 6. Thank you for this insight. We have made the necessary adjustment to avoid confusion to the reader. Specifically, we have adjusted BAMS to now denote “Bioadhesive Interface for Marine Sensors”. To avoid the potential confusion of the term “networks”, we have relocated this concept to the outlook section.. The underlying rationale for employing the idea of networks was rooted in the prospect that our proposed method could significantly increase the number of animals tagged in a short timeframe. Assuming an abundance of both sensors and animals. In this scenario, multiple nodes of data from sensors could be collected in a small timeframe, thus creating a network of data points. While the sensors in our study did not communicate with a central place, we believe the potential for further exploration of this aspect, which could be facilitated through the use of our technology.

Comment 7. I think the paper would benefit from being reorganized on the basis of its 3 parts. They should just follow the thought process that must have gone on. Firstly, mention the surgical glue (and reference that original paper), then discuss the general characteristics that make it effective in soft tissue repair. Then they should note that these characteristics are useful for an application in an entirely different realm, i.e. animal biotelemetry, where its use for attachment to soft bodied animals will make possible studies that are presently very difficult or unfeasible. Then description of the glue and its characteristics including its compliance and the testing of mechanical properties should follow in turn followed by, a description of the

laboratory tank testing on a variety of animals. This section should provide more information, including a clear description of the estimated how long the tags would stay on the animals. After all, this is the first question someone designing a sturdy using the glue would ask.

They say, “Due to the soft and robust characteristics of the adhesive interface, the “BAMS” can be reliably integrated with fragile aquatic animals, such as squid and jellyfish, for up to 3 days, allowing long-term animal behavior monitoring (Fig. 2k). Does this mean that 3 days is the maximum time the bond will last? How about the actual data on attachment time on the different animals. This is what practitioners will want to know, not the data on ODBA etc. In general, the data from the tags a is not the important part of the paper, only serving as an example of potential uses. Anyway, it is not presented in a biologically meaningful way. It is just an anecdotal account of what tags might record if used with this attachment method. I guess reporting the ODBA results shows that the mounting is not so compliant that it damps out the acceleration signal to a degree that obscures indications of animal activity but that is not the point that the text makes. It suggests different animals wiggle in different ways, not much more than that.

Figures 3 and 4 serve to show how tags might be used. The authors say in Fig 4. , “Group and community behavior for marine animals equipped with the BAMS. a, Schematic illustration of large-scale multi-species biologging experiment. Skate, squid, lobster, and flounder were placed in a large saltwater tank (318,100 liters), and roughly half of the individuals of each species were equipped with the BAMS”. Again, this is not really a biologging experiment to study animal behavior, rather it is an example of the sorts of animals on which that the attachment adhesive might be used. The value of the detailed data serve only that the attachment works. There is not natural community to study here.

And finally, Fig 5 provides a short description of the field test, and its value is primarily to show that the adhesive can be used in the field. It would have been useful to go into a bit more detail of exactly where, when and how the adhesive was prepared, stored and transported to where the animals were caught. The details of the data obtained during this test is of little relevance to the main value of the paper, which is purely the development and testing of the glue. The tag could just as well been a mock up. A single depth profile of light level and associated acceleration is of little significance in its own right. Here, the figure simply provides an example of how the attachment might be used.

Response 7. Thank you for your detailed comment regarding the movement data. We added details on adhesive preparation to improve methods clarity, and we presented requested tagging durations in figure 2k as well as in the text.

In response to your comment “they should note that these characteristics are useful for an application in an entirely different realm, i.e. animal biotelemetry, where its use for attachment to soft bodied animals will make possible studies that are presently very difficult or unfeasible... The value of the detailed data serve only that the attachment works. There is not natural community to study here.” – we would like to respectfully clarify the distinct biological importance of biologging and biotelemetry. While both biologging and biotelemetry involve the use of electronic devices to study animals, they serve different biological purposes. The

primary goal of biologging is to collect detailed, high-resolution data on an animal's activities, movements, and physiological parameters. This can include information such as location, diving depth, temperature, and other relevant environmental variables. Notably, these fine-scale behaviors and detailed environmental data are challenging to obtain through biotelemetry, which typically involves transmitting data over months at much lower temporal resolutions. Therefore, in figures 3, 4, and 5, we deliberately selected fine-scale behaviors that dominate these species' time and energy budgets, which have individual fitness and species competitive implications. Additionally, for squid, we show group-level schooling behaviors that are prevalent in reproductive aggregations, predator evasion, and migratory swimming. All of these behaviors have ecophysiological implications, and they are the kind of prominent behaviors that ecologists using biologging tags aim to quantify. Reduction of these sensing data would distract from the ultimate contribution of the paper – the presentation of the adhesive and its efficacy in measuring fine-scale behaviors through a reliable animal-sensor coupling. To summarize, depicting fine-scale animal behaviors via biologging are ecologically important and the animal-BIMS interface proves robust enough to capture and measure these behaviors accurately.

Comment 8. There should be more discussion of the mechanics of the bond as it decays over time due to biodegradation. The most important thing missing for me was missing for me was the detail on how long the tags would have stayed on of they were not actively removed and the state of the tissue after the attachment falls or is pulled off. In the tanks, were the tags actively removed or left on until they dropped off? I understand that there was an issue with maintenance of the squid in captivity, but how long were tags left attached? Were they actively removed? Were the mechanical properties of the glue changed by longer exposure to sea water? What did the scales or skin of the animals look like after forced or “eventual” detachment? These are the sort of details that I think the bio-logging community would like to hear more about so they could better understand what sorts of studies the glue would facilitate.

Response 8. Thank you for the insightful comment. Please refer to **Response 4** for relevant discussion on the adhesive removal. The adhesive interface allows a 3-day attachment on live squid, and we don't actively remove it. The squid looks healthy, exhibiting normal behavior after tag detachment. Tagged and tag-detached animals survived just as long as untagged animals. Note that biological factors in animals, such as secretions, sheltering habits, predator interactions etc., which are challenging to quantify, could also contribute to the variation of tag duration. Thanks a lot for your comments!

REVIEWERS' COMMENTS

Reviewer #4 (Remarks to the Author):

In my opinion that this version of the the paper is a stimulating and valuable contribution and worthy of publication.

During this iteration of the review process, I did not have available the previous version that had been submitted after the initial set of reviews, but I felt that this version read like a different and immeasurably improved paper. I have no significant reservations about its publication. You (the authors) seemed to have taken on board most of my suggestions and done a fundamental restructuring.

The only quibble I had was with this: “Based on our previous experimental results, the base remains on the animals after the release of ITAG for a certain time [ital. mine], and would eventually detach with the adhesive on it.”. This sounds rather vague. I guess this means that the adhesive was still strong enough to support the base but that you don’t really have a way of knowing how long it would have remained strong enough to support the whole tag assembly or even just the base because the animals were not recaptured. Ideally, I would have thought it informative and beneficial if the paper presented some hard data on how long the attachments last, if not on live animals in tanks, at least on models of some sort in sea water. I also was left wondering about how the strength of attachment changes over time in the marine environment. It would seem feasible to run such tests with the facilities they have available at WHOI or MIT. I think I would have just hung some weights on a attached plate and took note of when it pulled off or some other such simple test. I am left wondering about possible time course and consequences of bio-degradation. It happens that I am involved in a project involved in attaching tags to sharks using pins and this sort of data would have been very useful. And, as I mentioned in my previous review, flexible adhesives would be valuable for animals with scales, fur and feathers too.

What follows is not a comment on the paper but follows on from your response to my previous review where you said, “we would like to respectfully clarify the distinct biological importance of biologging and biotelemetry. While both biologging and biotelemetry involve the use of electronic devices to study animals, they serve different biological purposes. The primary goal of biologging is to collect detailed, high-resolution data on an animal's activities, movements, and physiological parameters. This can include information such as location, diving depth, temperature, and other relevant environmental variables. Notably, these fine-scale behaviors and detailed environmental data are challenging to obtain through biotelemetry, which typically involves transmitting data over months at much lower temporal resolutions. Therefore, in figures 3, 4, and 5, we deliberately selected fine-scale behaviors that dominate these species’ time and energy budgets, which have individual fitness and species competitive implications. Additionally, for squid, we show group-level schooling behaviors that are prevalent in reproductive aggregations, predator evasion, and migratory swimming. All of these behaviors have ecophysiological implications, and they are the kind of prominent behaviors that ecologists using biologging tags aim to quantify. Reduction of these sensing data would distract from the ultimate contribution of the paper – the presentation of the adhesive and its efficacy in measuring fine-scale

behaviors through a reliable animal-sensor coupling. To summarize, depicting fine-scale animal behaviors via biologging are ecologically important and the animal-BIMS interface proves robust enough to capture and measure these behaviors accurately

I would enjoy the chance to have a discussion with you about this. Since I am not likely to be passing by Cambridge or the Cape, the following will have to do.

This statement suggested to me that you have a somewhat different view of the biologging field and the extent of the literature than I do. It is an extremely broad church and the distinction they seem to be making between bio logging and bio-telemetry seems to suggest you underappreciate the complexity of the sorts of devices presently being deployed and the objectives of the studies in which they are used. There is a huge range of tag types being deployed in a rapidly expanding diversity of animals and a very large and active community engaged in doing the work (see : <https://bio-logging.net/>). Thi society defines biologging more broadly.

Any and all such devices face a range of constraints in terms of size, attachment limitations, energy requirements , sensor constraints, data storage and bandwidth, etc. Indeed, many devices now combine logging and transmission and then overcome any constraints using on-board data processing to get the best of both worlds. I.e. where high resolution acceleration data is both stored for subsequent possible recovery and also processed by the tag in real time to transmit analytical results. A good example of this is described in a recent paper where we describe methodology to get near real time info on the body condition of seals that allows us to determine if the animals are on a trajectory to successful reproduction. The tags capture fine scale, high resolution depth and acceleration data and then uses this to estimate buoyancy on-board so as to transmit the key information on the animal's body condition. (See Adachi et al. 2023 Body condition changes at sea: On-board calculation and telemetry of body density in diving animals; DOI: 10.1111/2041-210X.14089).

Comment 1:

They only quibble I had was with this: “Based on our previous experimental results, the base remains on the animals after the release of ITAG for a certain time [ital. mine], and would eventually detach with the adhesive on it.” This sounds rather vague. I guess this means that the adhesive was still strong enough to support the base but that you don’t really have a way of knowing how long it would have remained strong enough to support the whole tag assembly or even just the base because the animals were not recaptured. Ideally, I would have thought it informative and beneficial if the paper presented some hard data on how long the attachments last, if not on live animals in tanks, at least on models of some sort in sea water.

Response 1:

Thank you for the feedback. Our intention there is to highlight that, based on tank test observations, we anticipate the adhesive film to remain on the device (in this case the base) instead of lingering on the animal post-detachment. During the tank assessments, we consistently observed that the adhesive film remained on the device that had fallen off, not on the animal, and the device (sensor or mock sensor) remained attached for 1-3 days on squid in tanks. These tank experiments were carried out in UV-filtered sea water obtained directly from the nearby Vineyard Sound, MA.

Comment 2:

I also was left wondering about how the strength of attachment changes over time in the marine environment. It would seem feasible to run such tests with the facilities they have available at WHOI or MIT. I think I would have just hung some weights on a attached plate and took note of when it pulled off or some other such simple test. I am left wondering about possible time course and consequences of bio-degradation. It happens that I am involved in a project involved in attaching tags to sharks using pins and this sort of data would have been very useful. And, as I mentioned in my previous review, flexible adhesives would be valuable for animals with scales, fur and feathers too.

Response 2:

Regarding the adhesive strength over time in seawater, these tests were conducted using squid tissue in seawater over 24 hours at three different temperatures, which showed adhesive strength began to decrease after 24 hours (Supplementary Fig. 4). We found that leaving squid tissue soaking in seawater for longer than 24 hours led to tissue degradation and skin separation, which made it difficult to measure and distinguish the method of failure. The suggested tests hold significance in evaluating the adhesive-device interface, however, the weaker of the two interfaces is the tissue-adhesive one, as evidenced by the adhesive consistently remaining on the devices in every live animal test. While we focused our studies on soft fragile animals, using primarily squid as a testing platform, further work is required to investigate animals that have very different surface features and movement kinetics, both play a significant role in the forces experienced by the tag and how strong the tissue-adhesive interface is. The advantage of this method is that it's versatile, and can potentially be integrated into a variety of tagging methods, such as suction cups.

Comment 3:

What follows is not a comment on the paper but follows on from your response to my previous review where you said, “we would like to respectfully clarify the distinct biological importance of biologging and biotelemetry. While both biologging and biotelemetry involve the use of electronic devices to study animals, they serve different biological... I would enjoy the chance to have a discussion with you about this. Since I am not likely to be passing by Cambridge or the Cape, the following will have to do...”

Response 3:

Thank you for the clarification and links to your defining sources. In our response, we used the terms biotelemetry and biologging as defined in Hussey *et al* (2015). As it pertains to the present paper, the distinction is noted to tell the readers the sensors we used were extremely small and lightweight and did not contain transmitting acoustic beacon or satellite antenna. These components are relatively heavier than our sensor packages (ITAGs, and Technosmart Axy 5).

Regardless, we look forward to the potential opportunity to discuss your points in person as you suggest. Part of our team will be at the Biologging Science Symposium in Tokyo. We hope to connect there if you will be present.